# UniSim: A Unified Simulator for Time-Coarsened Dynamics of Biomolecules

**Ziyang Yu** [1]   **Wenbing Huang** [2 3]   **Yang Liu** [1 4]

## Abstract

Molecular Dynamics (MD) simulations are essential for understanding the atomic-level behavior of molecular systems, giving insights into their transitions and interactions. However, classical MD techniques are limited by the trade-off between accuracy and efficiency, while recent deep learning-based improvements have mostly focused on single-domain molecules, lacking transferability to unfamiliar molecular systems. Therefore, we propose **Uni**fied **Sim**ulator (UniSim), which leverages cross-domain knowledge to enhance the understanding of atomic interactions. First, we employ a multi-head pretraining approach to learn a unified atomic representation model from a large and diverse set of molecular data. Then, based on the stochastic interpolant framework, we learn the state transition patterns over long timesteps from MD trajectories, and introduce a force guidance module for rapidly adapting to different chemical environments. Our experiments demonstrate that UniSim achieves highly competitive performance across small molecules, peptides, and proteins.

## 1. Introduction

Molecular Dynamics (MD) simulations, an *in silico* method used to comprehend the time evolution of molecular systems in given environments, serve as a fundamental and essential tool in various fields like computational chemistry, pharmacology, material design, and condensed matter physics (Van Gunsteren & Berendsen, 1990; Lindorff-Larsen et al., 2011; Hollingsworth & Dror, 2018; Lau et al.,

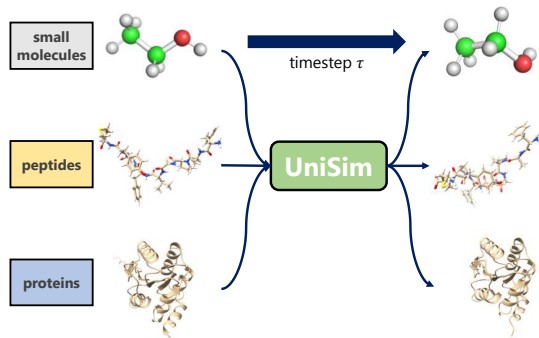

*Figure 1.* UniSim enables time-coarsened dynamics simulations of small molecules, peptides, and proteins over a long timestep $\tau$.

2018). One of the MD's core objectives is to generate trajectories of molecular states that adhere to underlying physics constraints over a period of time, given the initial state of the molecular system and environment configurations (Allen et al., 2004). To achieve this end, classical MD methods (van Gunsteren & Oostenbrink, 2024) update the next step of motion by numerically integrating Newton equations or Langevin dynamics (Langevin, 1908), with the potential energy and atomic forces calculated based on the current molecular state. It should be noted that the stability of the numerical integration requires an extremely small timestep $\Delta t \approx 10^{-15}$s during MD simulations (Plimpton, 1995).

In terms of how energy is calculated, classical MD methods can be divided into Quantum Mechanics (QM) methods (Griffiths & Schroeter, 2019) and empirical force field methods (Pearlman et al., 1995; Vanommeslaeghe et al., 2010). On the one hand, QM methods provide highly accurate energy calculations, while their great computational complexity makes it prohibitively expensive for accurate transition path sampling on a long time scale, like protein folding (Lindorff-Larsen et al., 2011). On the other hand, empirical force field methods are faster but less accurate. To accelerate the sampling of conformation transition pathways, some MD methods based on reinforcement learning (Shin et al., 2019), adaptive sampling (Markwick et al., 2011; Botu & Ramprasad, 2015a), as well as enhanced sampling (Bal & Neyts, 2015; M. Sultan & Pande, 2017; Wang et al., 2021; Célerse et al., 2022) have been proposed, yet achieving a

---

[*]Equal contribution [1]Department of Computer Science and Technology, Tsinghua University, Beijing, China [2]Gaoling School of Artificial intelligence, Renmin University of China, Beijing, China [3]Engineering Research Center of Next-Generation Intelligent Search and Recommendation, MOE [4]Institute for AI Industry Research (AIR), Tsinghua University, Beijing, China. Correspondence to: Wenbing Huang <hwenbing@126.com>, Yang Liu <liuyang2011@tsinghua.edu.cn>.

*Proceedings of the 42$^{nd}$ International Conference on Machine Learning*, Vancouver, Canada. PMLR 267, 2025. Copyright 2025 by the author(s).

qualitative improvement in efficiency while maintaining the accuracy remains challenging.

Recently, a surge of deep learning methods have been proposed to boost MD simulations from different aspects (Noé et al., 2019; Köhler et al., 2023; Lu et al., 2024; Klein et al., 2024; Schreiner et al., 2024; Wang et al., 2024a;b; Yu et al., 2024). Specifically, a series of methods categorized as *time-coarsened dynamics* aim to accelerate simulations by learning the push forward from $\vec{\mathbf{X}}_t$ to $\vec{\mathbf{X}}_{t+\tau}$ with a much larger timestep $\tau \gg \Delta t$, where $\vec{\mathbf{X}}_t$ denotes the molecular state at the wall-clock time $t$ (Klein et al., 2024; Schreiner et al., 2024; Yu et al., 2024). Although these methods can in principle achieve rapid long-time sampling, several issues arise in practical applications: 1) Almost all previous works are restricted to a single molecular domain (*e.g.*, peptides or proteins) within a fixed environment, lacking the transferability across different scenarios. 2) Some models leverage hand-crafted representations on specific domains (*e.g.*, $\gamma$-carbons in leucines), which significantly impair their ability to recognize unfamiliar molecules, such as proteins with unnatural amino acids (Link et al., 2003).

Taking all these into consideration, we propose a pretrained model for **Uni**fied full-atom time-coarsened dynamics **Sim**ulation (UniSim), which is transferable to small molecules, peptides as well as proteins, and can be easily adapted to various chemical environments by parameter-efficient training. An illustration for one-step simulation performed by UniSim is displayed in Figure 1. Firstly, owing to the scarcity of MD trajectory datasets, we propose to pretrain a unified atomic representation model on multi-domain 3D molecular datasets under different chemical environments and equilibrium states. Based on the pretrained model, we then leverage the *stochastic interpolants* generative framework (Albergo et al., 2023) to learn the push forward from $\vec{\mathbf{X}}_t$ to $\vec{\mathbf{X}}_{t+\tau}$, with a predefined long timestep $\tau$. Finally, in order to better adapt to specific chemical environments with different force conditions (temperatures, pressures, solvents etc.), we follow FBM (Yu et al., 2024) and introduce force guidance to regulate the sampling process of molecular trajectories. In summary, our contributions are:

- To our best knowledge, UniSim is the first deep learning-based generative model that tailored for transferable time-coarsened dynamics on cross-domain molecular systems.

- We employ a multi-task pretraining approach to learn a unified atomic representation model from cross-domain molecular data, leveraging novel techniques to tackle unbalanced molecular scales and provide fine-grained atomic representations.

- Based on the stochastic interpolant framework, we learn the state transition patterns over long-time steps

from MD trajectories, and introduce a force guidance module for rapidly adapting to different chemical environments.

## 2. Related Work

**3D Molecular Pretraining**  Given the success of the pretraining paradigm in the fields of NLP, some recent works have proposed pretraining methods based on 3D molecular structures (Zaidi et al., 2023; Luo et al., 2022; Zhang et al., 2023; Gao et al., 2022; Jiao et al., 2023; Zhou et al., 2023; Feng et al., 2024; Jiao et al., 2024; Zhang et al., 2024). Considering the scarcity of labeled molecular data, the NoisyNode (Zaidi et al., 2023) method proposes the denoising regularization paradigm for self-supervised pretraining on equilibrium conformations. Correspondingly, machine learning forcefields (Botu & Ramprasad, 2015b; Chmiela et al., 2017; 2018; 2023) explore the off-equilibrium conformation space by force-centric training. Furthermore, ET-OREO (Feng et al., 2024) proposes a novel pretraining method for learning unified representations encompassing both equilibrium and off-equilibrium conformations. EPT (Jiao et al., 2024) adopts the denoising pretraining on coarse-grained blocks and extends to learn representations of multi-domain molecules. Additionally, DPA-2 (Zhang et al., 2024) adapts to diverse chemical and materials systems by leveraging multi-task approach for different data sources. In this work, we propose a novel pretraining approach that is scalable to multi-domain biomolecules and adopts the multi-task approach to distinguish different forcefields and equilibrium states.

**Time-Coarsened Dynamics**  To overcome the constraints of numerical integration stability of classical MD simulations, a series of deep learning-based methods (Fu et al., 2023; Klein et al., 2024; Li et al., 2024; Jing et al., 2024; Schreiner et al., 2024; Hsu et al., 2024; Klein & Noe, 2024; Du et al., 2024; Yu et al., 2024) learn the push forward from $\vec{\mathbf{X}}_t$ to $\vec{\mathbf{X}}_{t+\tau}$ over long timesteps $\tau$ to enable rapid state transitions, which can be classified as *time-coarsened dynamics*. Various generative frameworks have been leveraged to learn the all-atom dynamics from data pairs of trajectories, including augmented normalizing flows (Klein et al., 2024), score-based diffusion (Hsu et al., 2024; Schreiner et al., 2024), flow matching (Klein & Noe, 2024) and stochastic interpolants (Du et al., 2024; Yu et al., 2024). On the contrary, F$^3$low (Li et al., 2024) and MDGen (Jing et al., 2024) learn dynamics of proteins with coarse-grained representations. Notably, FBM (Yu et al., 2024) introduces the force guidance to comply with the underlying Boltzmann distribution, achieving state-of-the-art performance on peptides. According to FBM, we first learn a unified vector field based on the stochastic interpolants framework, then we leverage the force guidance for parameter-efficient fine-tuning to

diverse chemical environments and forcefields.

# 3. Method

We present the overall workflow of UniSim in this section with an illustration in Figure 2. In § 3.1, we will define the task formulation with necessary notations. In § 3.2, we will propose novel techniques for obtaining a unified atomic representation model by pretraining on diverse datasets of multi-domain biomolecules. Afterwards, we will introduce the regimen of learning time-coarsened dynamics based on the stochastic interpolant in § 3.3. Finally, we will incorporate the force guidance technique to adapt to diverse chemical environments and present the scheme of parameter-efficient fine-tuning in § 3.4. All proofs of propositions are provided in § B.

## 3.1. Task Formulation

We assume that there are $m$ datasets $\{\mathcal{D}_1, \cdots, \mathcal{D}_m\}$ consisting of 3D molecular conformations and $l$ datasets $\{\mathcal{T}_1, \cdots, \mathcal{T}_l\}$ consisting of MD trajectories. Each molecule is represented as $\mathcal{G} = (\boldsymbol{Z}, \vec{\boldsymbol{X}})$, where $\boldsymbol{Z} \in \mathbb{R}^N$ denotes the atomic types of $N$ atoms, $\vec{\boldsymbol{X}} \in \mathbb{R}^{N \times 3}$ denotes the atomic positions. If the molecular conformation is off-equilibrium, we denote $\varepsilon : \mathbb{R}^{N \times 3} \to \mathbb{R}$ as the MD potential function and $\vec{\boldsymbol{F}} = -\nabla \varepsilon(\vec{\boldsymbol{X}}) \in \mathbb{R}^{N \times 3}$ as its MD forces, otherwise $\vec{\boldsymbol{F}} = \boldsymbol{0}$. Our goals are listed below:

1. We first obtain a unified atomic representation model $\varphi$ pretrained on the datasets $\{\mathcal{D}_1, \cdots, \mathcal{D}_m\}$. Formally, the pretrained model is defined as:

$$\boldsymbol{H}, \vec{\boldsymbol{V}} = \varphi(\bar{t}, \boldsymbol{Z}, \vec{\boldsymbol{X}}), \qquad (1)$$

   where $\bar{t} \in \{0, 1\}$ is set for the compatibility to the generation framework, $\boldsymbol{H} \in \mathbb{R}^{N \times H}$ denotes atomic representations of $H$ channels, and $\vec{\boldsymbol{V}} \in \mathbb{R}^{N \times 3 \times H}$ denotes SO(3)-equivariant vectors of all atoms.

2. Given the coarsened timestep $\tau$ and training data pairs $\{(\mathcal{G}_t, \mathcal{G}_{t+\tau})\}$ randomly sampled from a trajectory dataset $\mathcal{T}_i$ ($1 \leq i \leq l$), we train a vector field model $\phi$ with the pretrained $\varphi$ serving as a graph encoder, which learns the push forward from $\vec{\boldsymbol{X}}_t$ to $\vec{\boldsymbol{X}}_{t+\tau}$ based on the stochastic interpolant framework.

3. For adapting to diverse chemical environments (*e.g.*, solvation), we train a force guidance kernel $\zeta$ on the corresponding trajectory dataset $\mathcal{T}_j$ ($1 \leq j \leq l$), which incorporates underlying physics principles into generation (Yu et al., 2024). The parameters of networks $\varphi, \phi$ are kept frozen, serving as reusable backbones.

## 3.2. Unified Pretraining

In this section, we will introduce the techniques applied for pretraining UniSim on multi-domain 3D molecular datasets. Please note that, to comply with the symmetry properties of 3D molecules, we leverage the SO(3)-equivariant graph neural network `TorchMD-NET` (Pelaez et al., 2024) as the model architecture.

**Gradient-Environment Subgraph** A crucial challenge in training unified representation models arises from the vast scale discrepancy between molecular systems: small molecules typically contain tens of atoms, while proteins often comprise hundreds or thousands atoms. Most of previous works that construct molecular graphs using KNN (Kong et al., 2023) or radius cutoff (Schreiner et al., 2024) neglect the scale discrepancy, which may inhibit the transferability across molecular domains. To properly address the issue, we propose the so-called gradient-environment subgraph method to bridge the gap in scales of cross-domain molecules.

Specifically, for each macromolecule $\mathcal{G}$ with more than 1,000 atoms, we randomly select an atom $c$ with Cartesian coordinates $\vec{x}_c \in \mathbb{R}^3$. Given the predefined thresholds $0 \leq \delta_{\min} < \delta_{\max}$, the **gradient subgraph** $\mathcal{G}_g$ and the **environment subgraph** $\mathcal{G}_e$ are constructed as follows:

$$\mathcal{G}_g = \{j | j \in \mathcal{G}, ||\vec{x}_j - \vec{x}_c||_2 < \delta_{\min}\}, \qquad (2)$$
$$\mathcal{G}_e = \{j | j \in \mathcal{G}, ||\vec{x}_j - \vec{x}_c||_2 < \delta_{\max}\}. \qquad (3)$$

It is easy to show that $\mathcal{G}_g \subseteq \mathcal{G}_e$. Moreover, for each atom in $\mathcal{G}_g$, the distance to any other atom in $\mathcal{G} \setminus \mathcal{G}_e$ should be at least $\delta_{\max} - \delta_{\min}$. Therefore, as long as $\delta_{\max} - \delta_{\min}$ is set to be large enough, interactions between atoms outside of $\mathcal{G}_e$ and those in $\mathcal{G}_g$ can be neglected, which is consistent with physics principles.

Afterwards, the environment subgraph $\mathcal{G}_e$ rather than the whole graph $\mathcal{G}$ will serve as the input to the model. Edges between atoms are constructed based on a predefined cutoff $r_{\text{cut}}$. It should be noted that only the atoms within $\mathcal{G}_g$ participate in the calculation of the training objective, since the contribution of the truncated atoms of the original graph $\mathcal{G}$ to the atoms within $\mathcal{G}_e \setminus \mathcal{G}_g$ cannot be shielded.

**Atomic Embedding Expansion** In addition to the scale discrepancy, the molecular specificity is another key factor hindering the development of a unified atomic representation. In domains like proteins, atoms of the same type exhibit different but regular patterns (*e.g.*, CA and CB), while wet lab experiments (Rossmann & Arnold, 2001) elucidate that atoms of the same pattern probably share consistent properties like bond lengths. Due to the hard constraints on bond lengths and angles, these patterns exhibit discrete characteristics, which 3D GNNs tailored for continuous features

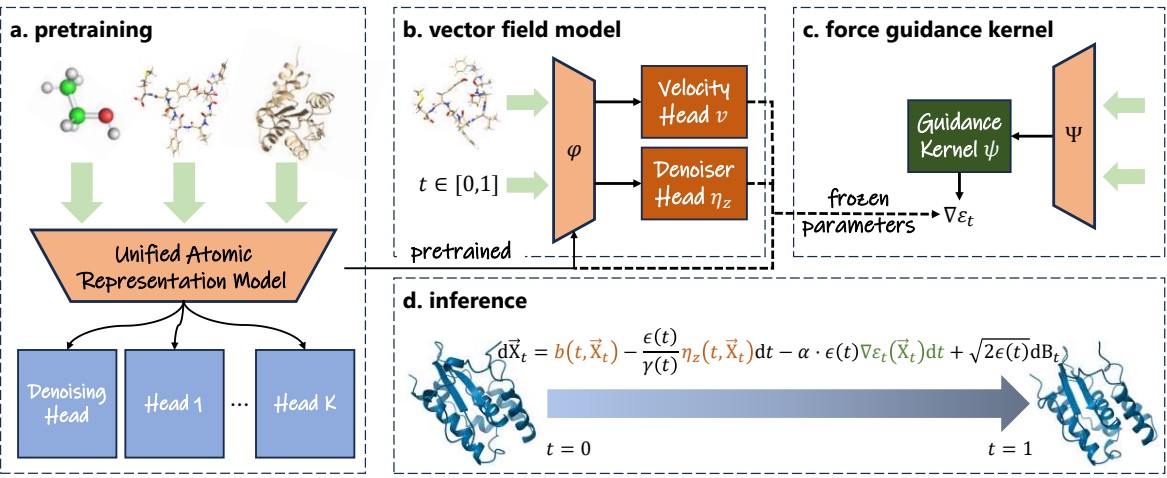

Figure 2. Illustration for the overall workflow of UniSim. **a.** The unified atomic representation model $\varphi$ is pretrained on multi-domain 3D molecules, where data from different chemical environments are fed to the corresponding output head. **b.** Based on the stochastic interpolant framework, vector field models $v, \eta_z$ are trained on MD trajectories to learn the push forward from $\vec{\mathbf{X}}_t$ to $\vec{\mathbf{X}}_{t+\tau}$ with timestep $\tau$. **c.** To adapt to different chemical environments, additional networks $\Psi, \psi$ are trained to fit the intermediate forcefield $\nabla \varepsilon_t$, with other parameters frozen. **d.** Given an initial state, inference is performed by iteratively solving an SDE with the diffusion time $t$ from 0 to 1.

may struggle to capture.

This implies that an effective embedding approach must capture these patterns. Using only the periodic table as vocabulary would yield low-resolution representations, missing those domain-specific regularities. Instead, we propose the atomic embedding expansion technique, extending elements of the periodic table to multiple discrete patterns that serve as the expanded vocabulary. Given the molecular graph, the model automatically maps each atom to the most possible pattern of the element based on its neighbors, thus simplifying the understanding of complex but highly regular structures.

Specifically, We first define a basic atomic vocabulary $\boldsymbol{A}_b \in \mathbb{R}^{A \times H}$ as well as an expanded atomic vocabulary $\boldsymbol{A}_e \in \mathbb{R}^{A \times D \times H}$ based on all possible element types that appear in the datasets, where $A$ represents the number of element types, $D$ represents the number of regular patterns for each element. Next, for any atom $i$ of the constructed molecular graph $\mathcal{G}$ and its neighbors $\mathcal{N}_i$, we compute the expanded weight vector $\boldsymbol{w}_i$ as follows:

$$\boldsymbol{n}_i = \sum_{j \in \mathcal{N}_i} \mathrm{rbf}(d_{ij}) \odot \boldsymbol{A}_b[j] \in \mathbb{R}^H, \qquad (4)$$

$$\boldsymbol{w}_i = \mathrm{softmax}\left(\mathrm{lin}(\boldsymbol{A}_b[i], \boldsymbol{n}_i)\right) \in [0,1]^D, \qquad (5)$$

where $\odot$ represents the element-wise multiplication operator, $\mathrm{lin}$ denotes the linear layer, $\mathrm{rbf}$ denotes the radial basis function, $\boldsymbol{V}_b[i]$ represents the atomic embedding of atom $i$, $d_{ij}$ denotes the Euclidean distance between atom $i$ and $j$. The vector $\boldsymbol{w}_i$ is further considered as the probability that atom $i$ appears in one of the $D$ possible regular chemical environments. To allow for back-propagation, we calculate

the **expanded embedding** of atom $i$ as follows:

$$\boldsymbol{z}_i = \mathrm{lin}(\boldsymbol{A}_b[i], \boldsymbol{w}_i^\top \boldsymbol{A}_e[i], \boldsymbol{n}_i) \in \mathbb{R}^H. \qquad (6)$$

Then $\boldsymbol{Z} \in \mathbb{R}^{N \times H}$ is the concatenation of the expanded embeddings $\boldsymbol{z}_i$ for all atoms $i$.

**Unified Multi-Head Pretraining**   With the graph topology and the atomic representation well prepared, we propose the technique for unified pretraining on multi-domain molecules. First of all, following Feng et al. (2024), all molecular data can be preliminarily classified into equilibrium and off-equilibrium categories. For the off-equilibrium molecular conformations $\vec{\boldsymbol{X}}$ with real MD forces $\vec{\boldsymbol{F}}$, we use the following objective for self-disciplined pretraining:

$$\mathcal{L}_o = ||\nabla_{\vec{\boldsymbol{X}}}(\sum_i \boldsymbol{H}_{\mathrm{out}}[i]) + \vec{\boldsymbol{F}}||_2^2, \qquad (7)$$

with

$$\boldsymbol{H}_{\mathrm{out}}, \vec{\boldsymbol{V}}_{\mathrm{out}} = \mathrm{GVP}(\boldsymbol{H}, \vec{\boldsymbol{V}}), \qquad (8)$$

where $\boldsymbol{H}, \vec{\boldsymbol{V}}$ are obtained by the representation model $\varphi$ based on Eq. (1), GVP is the Geometric Vector Perceptron (GVP) layer first introduced by Jing et al. (2021) and further updated by Jiao et al. (2024).

For the equilibrium molecular conformations, we adopt the denoising pretraining method. Specifically, we first add a Gaussian noise to the equilibrium conformation:

$$\vec{\boldsymbol{X}}' = C(\vec{\boldsymbol{X}} + \sigma_e \epsilon), \ \epsilon \sim \mathcal{N}(0, \boldsymbol{I}), \qquad (9)$$

where $C(\cdot)$ is the centering operator to ensure translational neutrality, $\sigma_e$ is the hyperparameter to control the noise

scale. Following Eqs. (1) and (8), we obtain $H'_{\text{out}} \in \mathbb{R}^N$ by using $\vec{X}'$ as the input. Then the training objective is given by:

$$\mathcal{L}_e = ||\nabla_{\vec{X}}(\sum_i \boldsymbol{H}'_{\text{out}}[i]) - \frac{\vec{X}' - \vec{X}}{\sigma_e}||_2^2. \qquad (10)$$

However, different datasets may have calculated MD forces under varying conditions (*e.g.*, solvation), leading to misalignment between data distributions. To address the issue, DPA-2 (Zhang et al., 2024) proposes the multi-head pretraining technique, introducing the unified descriptor for atomic representations and the fitting network that is updated exclusively with the specific pretraining dataset. Similarly, we use a series of GVP heads in Eq. (8) with non-shared weights to process data with force labels calculated in different force-fields. For convenience, we denote the training objective in Eq. (7) calculated by the $k$-th ($1 \leq k \leq K$) head as $\mathcal{L}_o^{(k)}$, where $K$ represents the number of forcefields across the pretraining datasets. The ultimate training objective for unified multi-head pretraining is given by:

$$\mathcal{L}_p = \mathcal{L}_e \mathbb{I}(\vec{F} = \mathbf{0}) + \mathcal{L}_o^{(k)} \mathbb{I}(\vec{F} \neq \mathbf{0}), \qquad (11)$$

where $\mathbb{I}(\cdot)$ is the indicator function, $k$ is the identifier for distinguishing different chemical environments.

### 3.3. Vector Field Model for Dynamics

In this section, we will present the generative framework of UniSim, named as the **vector field model**, to learn the push forward from $\vec{X}_t$ to $\vec{X}_{t+\tau}$ of MD trajectories with a predefined long timestep $\tau$. For compatibility to the generative framework, we will denote the initial state as $\vec{X}_0$ and the terminal state as $\vec{X}_1$ for each training data pair.

Here we leverage the *stochastic interpolant* (Albergo et al., 2023) as our generative framework. Given two probability density functions $q_0, q_1 : \mathbb{R}^d \to \mathbb{R}_{\geq 0}$, a stochastic interpolant between $q_0$ and $q_1$ is a stochastic process $\vec{X}_t$ defined as:

$$\vec{X}_t = I(t, \vec{X}_0, \vec{X}_1) + \gamma(t)\vec{Z}, \qquad (12)$$

where the pair $(\vec{X}_0, \vec{X}_1)$ is drawn from the probability measure $\nu$ that marginalizes on $q_0$ and $q_1$, $\vec{Z} \sim \mathcal{N}(0, \boldsymbol{I})$, $t \in [0, 1]$ denotes the diffusion time, $I(\cdot) : [0, 1] \times \mathbb{R}^d \times \mathbb{R}^d \to \mathbb{R}^d$ and $\gamma(\cdot) : [0, 1] \to \mathbb{R}$ are mapping functions that should satisfy certain properties.

To obtain training objectives, we first define the velocity $v$ and the denoiser $\eta_z$ as:

$$v(t, \vec{X}) = \mathbb{E}[\partial_t I(t, \vec{X}_0, \vec{X}_1)|\vec{X}_t = \vec{X}], \qquad (13)$$

$$\eta_z(t, \vec{X}) = \mathbb{E}[\vec{Z}|\vec{X}_t = \vec{X}], \qquad (14)$$

where the expectation is taken over the data pairs $(\vec{X}_0, \vec{X}_1)$. It can be further proved that the stochastic process $\vec{X}_t$, ve-

locity $v$ and denoiser $\eta_z$ are linked by an SDE:

$$d\vec{X}_t = b(t, \vec{X}_t)\, dt - \frac{\epsilon(t)}{\gamma(t)}\eta_z(t, \vec{X}_t)\, dt + \sqrt{2\epsilon(t)}\, d\mathbf{B}_t, \qquad (15)$$

where $b(t, \vec{X}) = v(t, \vec{X}) + \dot{\gamma}(t)\eta_z(t, \vec{X})$, $\epsilon(t)$ is a predefined function with regard to $t$, and $\mathbf{B}_t$ denotes the standard Wiener process.

Note that different choices of $I(\cdot)$ and $\gamma(\cdot)$ can induce infinite SDEs. As a well-studied case, we follow the setting of Yu et al. (2024):

$$I(t, \vec{X}_0, \vec{X}_1) = t\vec{X}_1 + (1 - t)\vec{X}_0, \gamma(t) = \sqrt{t(1 - t)}\sigma_s, \qquad (16)$$

where $\sigma_s \in \mathbb{R}_+$ is a hyperparameter to control the perturbation strength. Substituting them into Eqs. (13) and (14), the training objectives are given by:

$$\mathcal{L}_v = \mathbb{E}[||v(t, \vec{X}_t) - (\vec{X}_1 - \vec{X}_0)||_2^2 + ||\eta_z(t, \vec{X}_t) - \vec{Z}||_2^2], \qquad (17)$$

where $v$ and $\eta_z$ are implemented as neural networks, the expectation is taken over the diffusion time $t$ following the uniform distribution on $[0, 1]$, training data pairs $(\vec{X}_0, \vec{X}_1)$ and the intermediate state $\vec{X}_t$ following Eq. (12).

Here we introduce the implementation details of networks $v$ and $\eta_z$. First, the pretrained network $\varphi$ is initialized as a graph encoder, which now takes the diffusion time $t \in [0, 1]$ instead of $\bar{t} \in \{0, 1\}$ as input. The invariant and equivariant outputs of $\varphi$, $\boldsymbol{H}$ and $\vec{V}$, are further fed to two GVP layers with non-shared weights as the output heads of $v$ and $\eta_z$, respectively. Considering different scales of labels, an additional Vector LayerNorm (VLN) is stacked before each GVP layer, defined as as:

$$\text{VLN}(\vec{V}) = \vartheta \cdot \frac{\vec{V}}{\text{std}_1(\vec{V})} \in \mathbb{R}^{N \times 3 \times H}, \qquad (18)$$

where $\vartheta$ is a learnable parameter, $\text{std}_i(\cdot)$ represents calculating the standard value along dimension $i$ of the tensor.

### 3.4. Force Guidance Kernel for Finetuning

In this section, we will introduce the network $\zeta$, named as the **force guidance kernel**, which provides the mobility to different chemical environments by learning an "virtual" force defined on $\vec{X}_t$ and incorporating into generation. Inspired by Wang et al. (2024a); Yu et al. (2024), we hope the marginal distribution $p_t$ generated by $\zeta$ satisfies:

$$p_t(\cdot) \propto q_t(\cdot)\exp(-\alpha\varepsilon_t(\cdot)), \ \varepsilon_0(\cdot) = \varepsilon_1(\cdot) = \varepsilon(\cdot), \quad (19)$$

where $q_t$ represents the marginal distribution generated by the vector field model $\phi$, $\alpha$ is the hyperparameter of guidance strength, and $\varepsilon_t(\cdot)$ denotes an *intermediate potential*

defined on $\vec{\mathbf{X}}_t$, which is continuous with the MD potential $\varepsilon$ at two endpoints. Therefore, any change of MD potentials owing to different chemical environments will reflect on the change of generated marginal distributions based on $\zeta$, serving as the guidance of underlying physics principles.

For simplification, we assume that the stochastic interpolant that generates $p_t$ shares the same form as in Eq. (16), implying that $p_t(\vec{\mathbf{X}}_t|\vec{\mathbf{X}}_0, \vec{\mathbf{X}}_1) = q_t(\vec{\mathbf{X}}_t|\vec{\mathbf{X}}_0, \vec{\mathbf{X}}_1)$. According to Yu et al. (2024), the closed-form of the *intermediate forcefield* $\nabla \varepsilon_t(\cdot)$ is given by:

$$\nabla \varepsilon_t(\vec{\mathbf{X}}_t) = \frac{\mathbb{E}[q_t(\vec{\mathbf{X}}_t|\vec{\mathbf{X}}_0, \vec{\mathbf{X}}_1)g(\vec{\mathbf{X}}_0, \vec{\mathbf{X}}_1)h(\vec{\mathbf{X}}_0, \vec{\mathbf{X}}_1, \vec{\mathbf{X}}_t)]}{\alpha\mathbb{E}[q_t(\vec{\mathbf{X}}_t|\vec{\mathbf{X}}_0, \vec{\mathbf{X}}_1)g(\vec{\mathbf{X}}_0, \vec{\mathbf{X}}_1)]}, \quad (20)$$

where $g(\vec{\mathbf{X}}_0, \vec{\mathbf{X}}_1) = \exp(-\alpha(\varepsilon(\vec{\mathbf{X}}_0) + \varepsilon(\vec{\mathbf{X}}_1)))$ and $h(\vec{\mathbf{X}}_0, \vec{\mathbf{X}}_1, \vec{\mathbf{X}}_t) = \nabla \log q_t(\vec{\mathbf{X}}_t) - \nabla \log q_t(\vec{\mathbf{X}}_t|\vec{\mathbf{X}}_0, \vec{\mathbf{X}}_1)$. According to Albergo et al. (2023), $-\gamma^{-1}(t)\eta_z(t, \vec{\mathbf{X}}_t)$ is an unbiased estimation of $\nabla \log q_t(\vec{\mathbf{X}}_t)$, and $\nabla \log q_t(\vec{\mathbf{X}}_t|\vec{\mathbf{X}}_0, \vec{\mathbf{X}}_1)$ has a closed-form solution based on Eqs. (12) and (16), thus all terms of Eq. (20) can be calculated during training without any approximation.

Further, Proposition 3.1 reveals the closed-form of SDE that generates $p_t$:

**Proposition 3.1.** *Assume that marginals $q_t$ and $p_t$ are generated by $b(t, \vec{\mathbf{X}}), \eta_z(t, \vec{\mathbf{X}})$ and $b'(t, \vec{\mathbf{X}}), \eta_z'(t, \vec{\mathbf{X}})$ based on Eq. (15), respectively. Given the probability measure $\nu$ of data pairs satisfying $\nu(\vec{\mathbf{X}}_0, \vec{\mathbf{X}}_1) = \nu(\vec{\mathbf{X}}_1, \vec{\mathbf{X}}_0)$, we have the following equalities hold:*

$$b'(t, \vec{\mathbf{X}}) = b(t, \vec{\mathbf{X}}), \quad (21)$$

$$\eta_z'(t, \vec{\mathbf{X}}) = \eta_z(t, \vec{\mathbf{X}}) + \alpha\gamma(t)\nabla \varepsilon_t(\vec{\mathbf{X}}). \quad (22)$$

Finally, we introduce the implementation of the force guidance kernel $\zeta$. Keeping the parameters of the representation model $\varphi$ and the vector field model $\phi = \{v, \eta_z\}$ frozen, we use another `TorchMD-NET` $\Psi$ as the graph encoder of $\zeta$, which is initialized with the same hyperparameters of $\varphi$. To leverage the unified atomic representation, we adopt the residual mechanism, where the invariant output $H$ of $\varphi$ will be added to that of $\Psi$ during both training and inference.

After the invariant and equivariant features are obtained from $\Psi$, we construct the network $\psi$ to fit the intermediate forcefield $\nabla \varepsilon_t(\cdot)$ in the same interpolation form as in Yu et al. (2024):

$$\psi(\cdot) = (1-t)\psi_0(\cdot) + t\psi_1(\cdot) + t(1-t)\psi_2(\cdot), \quad (23)$$

where networks $\psi_i$ ($i \in \{0, 1, 2\}$) are implemented as the same architecture of $v$, and $\psi_0, \psi_1$ are used to fit MD force-fields at two endpoints $t = 0, 1$, respectively. Therefore, the

training objective of the force guidance kernel is given by:

$$\mathcal{L}_f = \mathbb{E}[||\psi_0(t, \vec{\mathbf{X}}_t) + \vec{\mathbf{F}}_0||_2^2 + ||\psi_1(t, \vec{\mathbf{X}}_t) + \vec{\mathbf{F}}_1||_2^2$$
$$+ ||\psi(t, \vec{\mathbf{X}}_t) - \frac{g(\vec{\mathbf{X}}_0, \vec{\mathbf{X}}_1)h(\vec{\mathbf{X}}_0, \vec{\mathbf{X}}_1, \vec{\mathbf{X}}_t)}{\alpha\mathbb{E}[q_t(\vec{\mathbf{X}}_t|\vec{\mathbf{X}}_0, \vec{\mathbf{X}}_1)g(\vec{\mathbf{X}}_0, \vec{\mathbf{X}}_1)]}||_2^2], \quad (24)$$

where $\vec{\mathbf{F}}_0, \vec{\mathbf{F}}_1$ correspond to MD forces of $\vec{\mathbf{X}}_0, \vec{\mathbf{X}}_1$, respectively, and the expectation in the denominator term is taken over all training data pairs of a mini-batch.

### 3.5. Inference

During inference, the SDE of Eq. (15) will be discretized into $T$ equidistant steps for generation, where $T$ is a hyperparameter. Given the initial state as $\vec{\mathbf{X}}_0$, a new state $\vec{\mathbf{X}}_1$ is generated through the $T$-step discrete Markov process, which completes one inference iteration. Subsequently, the newly generated state serves as the initial state for the next iteration, by which UniSim is able to autoregressively generate trajectories for any given chain length.

Empirically, we find that performing inference without post-processing leads to unstable conformation generation. Therefore, after each iteration, we add a conformation refinement step for peptides and proteins. Details and related analysis can be found in § C.2 and § E.2, respectively.

## 4. Experiments

### 4.1. Experimental Setup

**Datasets**   Firstly, the pretraining datasets $\mathcal{D}_i$ ($1 \leq i \leq m$) are listed as follows: 1) **PCQM4Mv2** (Hu et al., 2021), a quantum chemistry dataset with around 3M small molecules of equilibrium conformations optimized by DFT. 2) **ANI-1x** (Smith et al., 2020), a small organic molecule dataset consisting of 5M DFT calculations for QM properties like energies, atomic forces, etc. 3) **PepMD** (Yu et al., 2024), a peptide dataset including peptides of 3-10 residues with the sequence identity threshold of 60%, where we perform MD simulations using `OpenMM` (Eastman et al., 2017) to generate MD trajectories of 283 peptides. We adopt the same test set split of 14 peptides as in the original paper. 4) **Protein monomers** processed by Jiao et al. (2024), a subset of PDB (Berman et al., 2000) including protein monomer crystal structures. 5) **ATLAS** (Vander Meersche et al., 2024), a protein dataset gathering all-atom MD simulations of protein structures, which is chosen for structural diversity by ECOD domain classification (Schaeffer et al., 2017). Following Cheng et al. (2024), We selected proteins from the dataset of 2024.09.21 with no more than 500 residues and a coil percentage not exceeding 50%, resulting in a total of 834 data entries. We then apply the sequence clustering with the threshold of 30% by `MMseq2` (Steinegger & Söding, 2017), obtaining 790/14 as the train/test splits. 6) **Solvated Protein**

**Fragments** (SPF) (Unke & Meuwly, 2019), a dataset probing many-body intermolecular interactions between protein fragments and water molecules. We randomly split the above pretraining datasets for training and validation by 4:1.

Secondly, we will validate the transferability of UniSim across three molecular domains: 1) **MD17** (Chmiela et al., 2017) and **MD22** (Chmiela et al., 2023) used for training and evaluation respectively, serving as representatives for small molecules. 2) **PepMD** as introduced above for peptides and 3) **ATLAS** for proteins. More details of pretraining datasets, trajectory datasets and MD simulation setups are shown in § D.

**Baselines**  Considering computational resource limitations, we compare UniSim with baseline models on peptides and with MD trajectories on other molecular domains. Specifically, we select the following deep learning-based models as our baselines: 1) FBM (Yu et al., 2024), the current state-of-the-art model based on bridge matching, learning time-coarsened dynamics on peptides with steerable force guidance. 2) Timewarp (Klein et al., 2024), a generative model leveraging the augmented normalizing flow and MCMC techniques, exhibiting transferability to small peptide systems. 3) ITO (Schreiner et al., 2024), a conditional diffusion model tailored for learning dynamics on varying time resolutions. 4) Score Dynamics (SD) (Hsu et al., 2024), a score matching diffusion model that captures transitions of collective variables of interest.

**Metrics**  We employ the same metrics in Yu et al. (2024) to comprehensively evaluate the distributional similarity, validity and flexibility of generated ensembles. Briefly, the metrics are listed below: 1) The proportion of conformations with neither bond break nor bond clash, termed as VAL-CA (Lu et al., 2024). 2) The root mean square error of contact maps between generated ensembles and MD trajectories, termed as CONTACT (Janson et al., 2023). 3) The Jensen-Shannon (JS) distance on projected feature spaces of Pairwise Distances (PWD), Radius-of-Gyration (RG), the slowest two Time-lagged Independent Components (TIC) (Pérez-Hernández et al., 2013) as well as their joint distribution, termed as TIC-2D. The mean value of JS distances along each dimension is reported.

## 4.2. Evaluation on Peptides

Based on the unified atomic representation model $\varphi$ obtained through pretraining, we train on PepMD to derive a unified vector field model $\phi$. The rationale behind the choice is that peptides have moderate scales and exhibit high structural flexibility, making it suitable for transferring to other molecular domains subsequently. For fair comparison, all baselines are trained from scratch on PepMD until convergence, and sample trajectories of each test peptide for

a chain length of $10^3$. The evaluation results on all 14 test peptides of PepMD are shown in Table 1, where UniSim/g refers to the version that performs inference using only the vector field model $\phi$ without force guidance.

Based on Table 1, we first observe that UniSim outperforms the baselines on nearly all metrics, particularly when compared to FBM that uses a similar framework, demonstrating the effectiveness of our pretraining techniques. Moreover, by introducing the force guidance, UniSim shows a significant improvement in validity while maintaining the same level of distribution similarity, which reveals a deeper comprehension of the underlying physics constraints.

*Table 1.* Results on the test set of PepMD. Values of each metric are shown in mean/std of all 14 test peptides. The best result for each metric is shown in **bold** and the second best is underlined.

| MODELS | JS DISTANCE (↓) | | | | VAL-CA (↑) | CONTACT (↓) |
|---|---|---|---|---|---|---|
| | PWD | RG | TIC | TIC-2D | | |
| FBM | 0.361/0.165 | 0.411/0.224 | 0.510/0.124 | 0.736/0.065 | 0.539/0.111 | 0.205/0.105 |
| TIMEWARP | 0.362/0.095 | 0.386/0.120 | 0.514/0.110 | 0.745/0.061 | 0.028/0.020 | 0.195/0.051 |
| ITO | 0.367/0.077 | 0.371/0.131 | **0.495**/0.126 | 0.748/0.055 | 0.160/0.186 | 0.174/0.099 |
| SD | 0.727/0.089 | 0.776/0.087 | 0.541/0.113 | 0.782/0.042 | 0.268/0.266 | 0.466/0.166 |
| UniSim/g | 0.332/0.135 | 0.332/0.161 | 0.510/0.115 | 0.738/0.064 | 0.505/0.112 | 0.162/0.076 |
| UniSim | **0.328**/0.149 | **0.330**/0.189 | 0.510/0.124 | **0.731**/0.074 | **0.575**/0.139 | **0.157**/0.088 |

For a more intuitive understanding, we provide the visualization of the metrics for the two test cases in Figure 3. UniSim exhibits a close alignment with MD trajectories with regard to pairwise distances and residue contact rates. Furthermore, UniSim shows a good recovery of known metastable states, with samples located mainly in high-density regions. Though fully reproducing the free energy landscape may require longer trajectory lengths, UniSim still demonstrates a basic understanding of the intrinsic Boltzmann distribution.

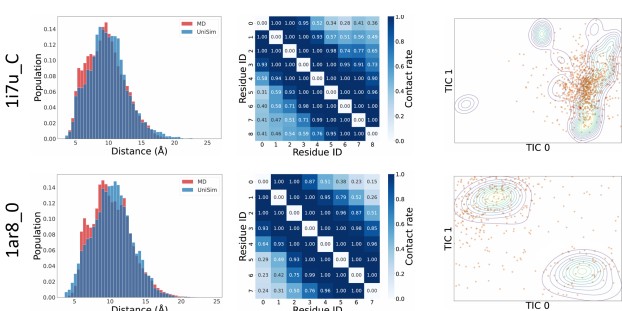

*Figure 3.* The visualization of comprehensive metrics on peptide 1i7u_C (upper) and 1ar8_0 (lower). The left column shows the joint distribution of pairwise distances. The middle column demonstrates the residue contact map, where data in the lower and upper triangle are obtained from UniSim and MD, respectively. The right column displays TIC-2D plots for the slowest two components, where contours indicate the kernel density estimated on MD trajectories and the generated conformations are shown in scatter.

### 4.3. Transferability to Small Molecules

We further investigate the performance of UniSim on small molecules. Keeping parameters of the atomic representation model $\varphi$ and the vector field model $\phi$ frozen, we train on MD17 to obtain a force guidance kernel $\zeta$ and evaluate on 5 organic molecules of MD22. Here we use the relative distances between all heavy atom pairs as the projected feature to compute TIC and TIC-2D metrics. The ablation results comparing UniSim with UniSim/g are shown in Table 2. Apparently, the force guidance kernel helps the model transfer to a new chemical environment with better distributional similarity in overall.

*Table 2.* Results on the test set of MD22. Values of each metric are shown in mean/std of all 5 test molecules. The best result for each metric is shown in **bold**.

| MODELS | JS DISTANCE ($\downarrow$) | |
| --- | --- | --- |
| | TIC | TIC-2D |
| UniSim/g | 0.408/0.111 | 0.791/0.044 |
| UniSim | **0.368**/0.132 | **0.765**/0.063 |

To better understand how the force guidance kernel works, we provide visualizations on Ac-Ala3-NHMe and DHA in Figure 4. It shows that the force guidance greatly helps comprehend the free energy landscape, enabling more accurate transitions between metastable states. Accordingly, samples generated by UniSim are more likely concentrated in high-density regions, which complies with physics constraints of the specific chemical environment.

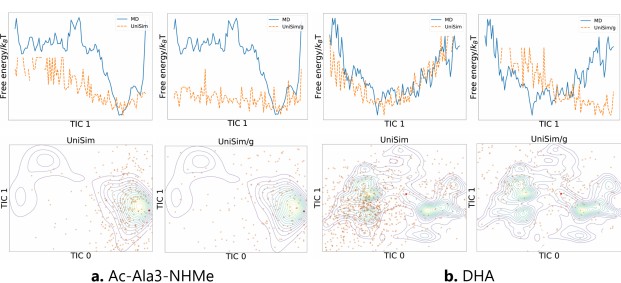

**a.** Ac-Ala3-NHMe  **b.** DHA

*Figure 4.* TIC and TIC-2D plots of UniSim (left) and UniSim/g (right) on **a.** Ac-Ala3-NHMe and **b.** DHA. The first row displays the free energy projection on TIC 1, and the second row demonstrates TIC plots for the slowest two components.

### 4.4. Exploration of Proteins

In this section, we explore the transferability of UniSim to proteins. Considering the complexity and specificity of protein structures, we first finetune the vector field model $\phi$ on ATLAS with a learning rate of 1e-4. Next, the force

guidance kernel corresponding to the protein domain is trained on ATLAS subsequently.

Afterwards, the models are evaluated on the test set of ATLAS and the results are displayed in Table 3. Evidently, UniSim significantly outperforms the baselines across all metrics, especially in terms of validity, achieving an improvement from 5% (ITO) to 8%. Meanwhile, the introduction of force guidance leads to modest improvements over most metrics as well.

*Table 3.* Results on the test set of ATLAS. Values of each metric are shown in mean/std of all 14 test protein monomers. The best result for each metric is shown in **bold**.

| MODELS | JS DISTANCE ($\downarrow$) | | | VAL-CA ($\uparrow$) | CONTACT ($\downarrow$) |
| --- | --- | --- | --- | --- | --- |
| | PWD | RG | TIC | | |
| FBM | 0.519/0.023 | 0.597/0.121 | 0.621/0.152 | 0.012/0.007 | 0.252/0.039 |
| ITO | 0.588/0.027 | 0.775/0.042 | 0.624/0.121 | 0.052/0.008 | 0.428/0.020 |
| SD | 0.604/0.020 | 0.762/0.060 | 0.605/0.128 | 0.001/0.000 | 0.235/0.033 |
| UniSim/g | 0.508/0.021 | 0.569/0.146 | 0.543/0.141 | 0.071/0.029 | **0.171**/0.031 |
| UniSim | **0.506**/0.021 | **0.554**/0.149 | **0.542**/0.159 | **0.079**/0.033 | 0.173/0.031 |

Moreover, to intuitively demonstrate the simulation efficiency of UniSim on large proteins, we present the TIC-2D visualizations of the first 200 generated conformations for two protein test cases, as shown in Figure 5. Notably, although long-term stable simulations for large proteins still proves to be challenging, UniSim is able to cross the energy barrier and explore distinct metastable states with only a few inference iterations, which shows the potential to take the place of traditional MD in terms of efficiency.

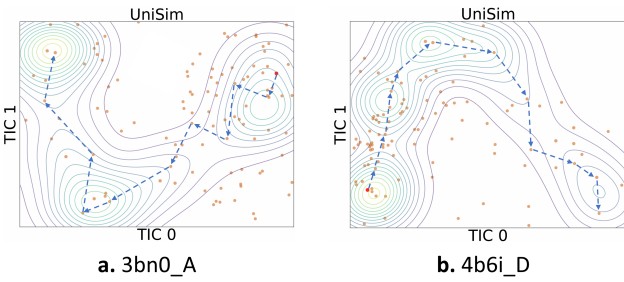

**a.** 3bn0_A  **b.** 4b6i_D

*Figure 5.* TIC-2D plots of the first 200 generated conformations for **a.** 3bn0_A and **b.** 4b6i_D. Contours indicate the kernel density estimated on MD trajectories, the generated conformations are shown in scatter, and the blue dashed arrows represent the order in which the conformations are generated.

### 4.5. Case Study: Alanine-Dipeptide

To further validate the stability and applicability of UniSim in long-timescale simulations, we conduct additional experiments on a well-studied molecular system, alanine-dipeptide (AD), consisting of only 22 atoms while exhibiting comprehensive free energy landscapes.

Specifically, following the task setup in Timewarp (Klein et al., 2024), we attempt to finetune UniSim trained on PepMD to AD before performing long-timescale simulations. Firstly, we obtain three independently sampled MD trajectories of AD with the simulation time of 250 ns from `mdshare`[1], which are assigned as the training/validation/test trajectories. The coarsened timestep $\tau$ is set to 100 ps, with 200,000 data pairs randomly sampled for training and validation from corresponding trajectories, respectively. UniSim is then finetuned on the curated AD dataset with the learning rate of 1e-4 for 300 epochs.

After we obtain the best checkpoint of UniSim evaluated on the validation set, we perform long-timescale simulations for a chain length of 100,000 to explore the metastable states of AD. We show the Ramachandran and TIC-2D plots of UniSim and the test MD trajectory in Figure 6. Building upon previous research (Wang et al., 2014), UniSim has demonstrated robust performance in long-timescale simulations by effectively exploring key metastable states of AD, including C5, $C7_{eq}$, $\alpha_R'$, $\alpha_R$ as well as $\alpha_L$. Moreover, the relative weights of generated conformation ensembles across different metastable states show good agreement with MD, indicating that UniSim is basically capable of reproducing the free energy landscape.

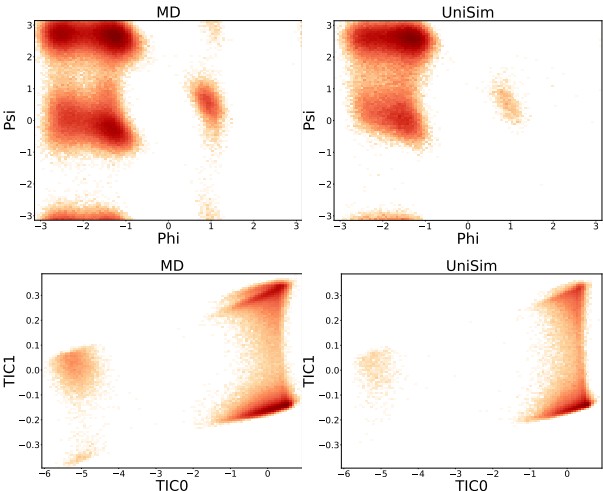

Figure 6. Visualization of Ramachandran plots (the first row) and TIC-2D plots (the second row) of UniSim and MD on alanine-dipeptide.

Furthermore, we provide a more intuitive illustration of the accuracy in generating conformations of metastable states. Based on Ramachandran plots, we apply the K-means (MacQueen, 1967) algorithm to obtain 5 clusters from MD trajectories, and select the centroid of each cluster as the representative conformation of the corresponding metastable state. Subsequently, we identify the conformation with the lowest

[1]https://markovmodel.github.io/mdshare/ALA2/#alanine-dipeptide

root-mean-square deviation (RMSD) to each representative conformation from trajectories generated by UniSim, which are illustrated in Figure 7. It can be observed that UniSim consistently exhibits an excellent recovery of metastable states with negligible deviation.

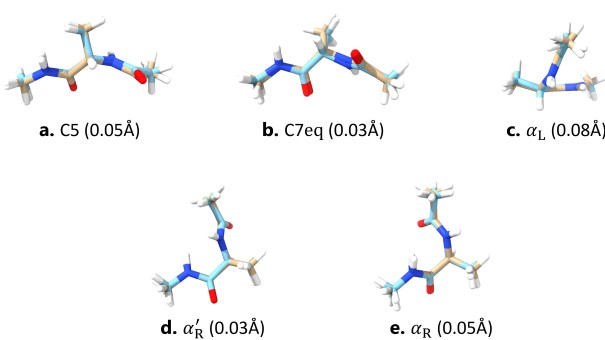

Figure 7. Comparison of representative conformations of AD between UniSim (blue) and MD (yellow), including: **a.** C5, **b.** $C7_{eq}$, **c.** $\alpha_L$, **d.** $\alpha_R'$ and **e.** $\alpha_R$. The RMSD values of each representative pair over heavy atoms are shown in brackets.

## 5. Conclusion and Future Work

In this work, we present a novel architecture called UniSim, the first deep learning-based model tailored for performing time-coarsened dynamics on biomolecules of diverse domains. To accommodate molecular inputs of different sizes and types, we obtain a unified atomic representation model by pretraining on multi-domain 3D molecular datasets with novel techniques. Afterwards, we leverage the stochastic interpolant framework to construct a vector field model that learns time-coarsened dynamics on MD trajectories. Considering the impact of different chemical environments in applications, we further introduce the force guidance kernel, which adjusts the expectation of observables by incorporating "virtual" forces into the stochastic process. Experiments conducted on small molecules, peptides and proteins have fully verified the superiority of UniSim in distribution similarity compared to MD trajectories and transferability to out-of-distribution molecular domains.

In addition, we believe this work can be advanced in the following aspects: 1) Influenced by cumulative prediction errors, the validity of the samples generated by UniSim is not fully reliable, especially for macromolecules like proteins. Efficient cross-domain structure optimization deserves further exploration. 2) The generated trajectories for evaluation in our experiment are relatively short, which may hinder the model from discovering more possible metastable states. The dynamics pattern of biomolecules over longer time scales is worth investigating.

## Acknowledgements

This work is jointly supported by the National Key R&D Program of China (No.2022ZD0160502), the National Natural Science Foundation of China (No. 61925601, No. 62376276, No. 62276152), and Beijing Nova Program (20230484278).

## Impact Statement

This paper presents work whose goal is to advance the field of molecular dynamics through developing a novel deep learning-based generative model, UniSim, which proposes a solution to the key limitation in current methods of unified and efficient simulation for cross-domain molecules. Our work has the potential to impact both scientific research and practical applications in real-world scenarios, such as drug discovery and molecular property prediction. We wish our work could inspire future research in this field.

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

## A. Reproducibility

Our code is available at https://github.com/yaledeus/UniSim.

## B. Proofs of Propositions

*Proof of Proposition 3.1.* Firstly, denote the marginal score of the vector field model $\phi$ as $s(t, \vec{\mathbf{X}})$, which is given by:

$$s(t, \vec{\mathbf{X}}) = \mathbb{E}[\nabla \log q_t(\vec{\mathbf{X}}|\vec{\mathbf{X}}_0, \vec{\mathbf{X}}_1)|\vec{\mathbf{X}}_t = \vec{\mathbf{X}}] \tag{25}$$

$$= \iint \nabla \log q_t(\vec{\mathbf{X}}|\vec{\mathbf{X}}_0, \vec{\mathbf{X}}_1) q_t(\vec{\mathbf{X}}_0, \vec{\mathbf{X}}_1|\vec{\mathbf{X}}) \, d\vec{\mathbf{X}}_0 \, d\vec{\mathbf{X}}_1 \tag{26}$$

$$= \frac{1}{q_t(\vec{\mathbf{X}})} \iint \nabla q_t(\vec{\mathbf{X}}|\vec{\mathbf{X}}_0, \vec{\mathbf{X}}_1) q(\vec{\mathbf{X}}_0, \vec{\mathbf{X}}_1) \, d\vec{\mathbf{X}}_0 \, d\vec{\mathbf{X}}_1 \tag{27}$$

$$= \frac{1}{q_t(\vec{\mathbf{X}})} \nabla \iint q_t(\vec{\mathbf{X}}|\vec{\mathbf{X}}_0, \vec{\mathbf{X}}_1) q(\vec{\mathbf{X}}_0, \vec{\mathbf{X}}_1) \, d\vec{\mathbf{X}}_0 \, d\vec{\mathbf{X}}_1 \tag{28}$$

$$= \frac{\nabla q_t(\vec{\mathbf{X}})}{q_t(\vec{\mathbf{X}})} = \nabla \log q_t(\vec{\mathbf{X}}), \tag{29}$$

where in the second equality we use the Bayesian rule, and the third equality is justified by assuming the integrands satisfy the regularity conditions of the Leibniz Rule. $q(\vec{\mathbf{X}}_0, \vec{\mathbf{X}}_1)$ denotes the joint distribution of training data pairs.

Further, according to Albergo et al. (2023), we have the following inference:

$$s(t, \vec{\mathbf{X}}) = -\gamma^{-1}(t)\eta_z(t, \vec{\mathbf{X}}). \tag{30}$$

Based on the assumption of Eq. (19), we have:

$$\nabla \log p_t(\vec{\mathbf{X}}) = \nabla \log q_t(\vec{\mathbf{X}}) - \alpha \nabla \varepsilon_t(\vec{\mathbf{X}}). \tag{31}$$

According to the above derivation, by replacing $\nabla \log p_t(\vec{\mathbf{X}}), \nabla \log q_t(\vec{\mathbf{X}})$ with $-\gamma^{-1}(t)\eta'_z(t, \vec{\mathbf{X}}), -\gamma^{-1}(t)\eta_z(t, \vec{\mathbf{X}})$, respectively, we show that $\eta'_z(t, \vec{\mathbf{X}}) = \eta_z(t, \vec{\mathbf{X}}) + \alpha\gamma(t)\nabla \varepsilon_t(\vec{\mathbf{X}})$ holds.

Secondly, we denote the following two terms on the probability path $q_t$:

$$u_f(t, \vec{\mathbf{X}}) = \mathbb{E}[\frac{\vec{\mathbf{X}}_1 - \vec{\mathbf{X}}}{1 - t}|\vec{\mathbf{X}}_t = \vec{\mathbf{X}}], \ u_r(t, \vec{\mathbf{X}}) = \mathbb{E}[\frac{\vec{\mathbf{X}} - \vec{\mathbf{X}}_0}{t}|\vec{\mathbf{X}}_t = \vec{\mathbf{X}}]. \tag{32}$$

Similarly, we define $u'_f(t, \vec{\mathbf{X}})$ and $u'_r(t, \vec{\mathbf{X}})$ on the probability path $p_t$ in the same form. Based on Eqs. (13), (14) and (16), we have:

$$v(t, \vec{\mathbf{X}}) = \mathbb{E}[\partial_t I(t, \vec{\mathbf{X}}_0, \vec{\mathbf{X}}_1)|\vec{\mathbf{X}}_t = \vec{\mathbf{X}}] = \mathbb{E}[\vec{\mathbf{X}}_1 - \vec{\mathbf{X}}_0|\vec{\mathbf{X}}_t = \vec{\mathbf{X}}], \tag{33}$$

$$\eta_z(t, \vec{\mathbf{X}}) = \mathbb{E}[\vec{\mathbf{Z}}|\vec{\mathbf{X}}_t = \vec{\mathbf{X}}] = \mathbb{E}[\frac{\vec{\mathbf{X}} - t\vec{\mathbf{X}}_1 - (1 - t)\vec{\mathbf{X}}_0}{\sqrt{t(1 - t)}\sigma_s}|\vec{\mathbf{X}}_t = \vec{\mathbf{X}}]. \tag{34}$$

Combining the connection of $b(t, \vec{\mathbf{X}}) = v(t, \vec{\mathbf{X}}) + \dot{\gamma}(t)\eta_z(t, \vec{\mathbf{X}})$, we have:

$$b(t, \vec{\mathbf{X}}) = \mathbb{E}[\vec{\mathbf{X}}_1 - \vec{\mathbf{X}}_0 + \frac{(1 - 2t)\sigma_s}{2\sqrt{t(1 - t)}} \cdot \frac{\vec{\mathbf{X}} - t\vec{\mathbf{X}}_1 - (1 - t)\vec{\mathbf{X}}_0}{\sqrt{t(1 - t)}\sigma_s}|\vec{\mathbf{X}}_t = \vec{\mathbf{X}}] \tag{35}$$

$$= \mathbb{E}[\frac{(1 - 2t)\vec{\mathbf{X}} + t\vec{\mathbf{X}}_1 - (1 - t)\vec{\mathbf{X}}_0}{2\sqrt{t(1 - t)}}|\vec{\mathbf{X}}_t = \vec{\mathbf{X}}] \tag{36}$$

$$= \frac{1}{2}(u_f(t, \vec{\mathbf{X}}) + u_r(t, \vec{\mathbf{X}})). \tag{37}$$

Similarly, $b'(t, \vec{\mathbf{X}}) = \frac{1}{2}(u'_f(t, \vec{\mathbf{X}}) + u'_r(t, \vec{\mathbf{X}}))$. According to Yu et al. (2024), it has been proven that $u'_r(t, \vec{\mathbf{X}}) - u_r(t, \vec{\mathbf{X}}) = u_f(t, \vec{\mathbf{X}}) - u'_f(t, \vec{\mathbf{X}})$ given the probability measure $\nu$ of training data pairs satisfying $\nu(\vec{\mathbf{X}}_0, \vec{\mathbf{X}}_1) = \nu(\vec{\mathbf{X}}_1, \vec{\mathbf{X}}_0)$. Therefore, we can derive that $b'(t, \vec{\mathbf{X}}) = b(t, \vec{\mathbf{X}})$. $\square$

## C. Training and Inference Details

### C.1. Normalization of Potential and Force Labels

Previous works Wang et al. (2024a); Yu et al. (2024) have mentioned the numerical instability of potential energies across different molecular systems. To ensure the stability of the training process, the potential labels are normalized during pre-processing. Specifically, For each trajectory of one molecular system, we denote the concatenation of unnormalized potential labels of all conformations as $\boldsymbol{E} \in \mathbb{R}^M$, where $M$ represents the trajectory length. The normalized potentials are given by:

$$\hat{\boldsymbol{E}} = \frac{\boldsymbol{E} - \max(\boldsymbol{E})}{\max(\boldsymbol{E}) - \min(\boldsymbol{E})} \in [-1, 0]^M, \tag{38}$$

where $\max(\cdot)$ and $\min(\cdot)$ denote the maximum and minimum element of the vector. Based on the technique, all potential energy labels are normalized to $[-1, 0]$ without manipulating the energy distribution of each ensemble.

For atomic force labels which are comparably more stable, we unify them into `MJ/(mol·nm)`, whose scale is close to standard Gaussian distributions empirically.

### C.2. Conformation Refinement

In practice, we found that small prediction errors in the autoregressive generation process would accumulate over time, resulting in unreasonable conformations especially for proteins. Following Wang et al. (2024a); Yu et al. (2024), after generating a new conformation, we introduce an energy minimization step using `OpenMM` (Eastman et al., 2017) with harmonic constraints for local structure refinement. Specifically, independent harmonic constraints are first added on all heavy atoms with spring constant of 10 kcal/mol·Å$^2$ in case of altering the overall conformation. The minimization tolerance is set to be 2.39 kcal/mol·Å$^2$ without maximal step limits. Afterwards, the energy minimization step will be applied for peptides or proteins, while small molecules will not undergo any additional treatment.

### C.3. Hyperparameters

The hyperparameters of UniSim for model constructing, training and inference are shown in Table 4.

## D. Experimental Details

### D.1. Dataset Details

**Dataset Curation** All peptides of PepMD are simulated by `OpenMM` (Eastman et al., 2017) with the implicit solvation model `GB-OBC I` (Onufriev et al., 2004), where we show MD simulation setups in Table 5. Moreover, we use the same forcefield to obtain MD potential and force labels of protein conformations in ATLAS. The pretraining dataset details are shown in Table 6, and we summarize the trajectory datasets in Table 7.

**Test Set** Molecules of each test set are listed below, where peptides or protein monomers are named as "`{PDB_id}_{chain_id}`":

- **MD22**: AT-AT-CG-CG, AT-AT, DHA, stachyose, Ac-Ala3-NHMe.

- **PepMD**: 1hhg_C, 1k8d_P, 1k83_M, 1bz9_C, 1i7u_C, 1gxc_B, 1ar8_0, 2xa7_P, 1e28_C, 1gy3_F, 1n73_I, 1fpr_B, 1aze_B, 1qj6_I.

- **ATLAS**: 1chd_A, 2nnu_A, 2bjq_A, 2ov9_C, 1r9f_A, 7rm7_A, 4adn_B, 3bn0_A, 3eye_A, 1h2e_A, 1jq5_A, 4b6i_D, 2znr_A, 1huf_A.

## E. Additional Experimental Results

### E.1. Ablation Study

**Hyperparameter Sensitivity** In order to investigate the hyperparameter sensitivity, we conduct ablation studies of SDE steps $T$ and the guidance strength $\alpha$ on PepMD test set, where the ablation results are shown in Table 8. From the table, we

*Table 4.* Hyperparameters of UniSim.

| Hyperparameters | Values |
|---|---|
| **Model** | |
| Hidden dimension $H$ | 256 |
| FFN dimension | 512 |
| RBF dimension | 64 |
| Expand embed dimension $D$ | 32 |
| # attention heads | 8 |
| # layers of atomic representation model | 4 |
| # layers of force guidance kernel | 4 |
| Gradient subgraph threshold $\delta_{\min}$ | 8Å |
| Environment subgraph threshold $\delta_{\max}$ | 20Å |
| Cutoff threshold $r_{\mathrm{cut}}$ | 5Å |
| Pretraining noise scale $\sigma_e$ | 0.04 |
| SDE perturbation strength $\sigma_s$ | 0.2 |
| **Training** | |
| Learning rate | 5e-4 |
| Optimizer | Adam |
| Warm up steps | 1,000 |
| Warm up scheduler | LamdaLR |
| Training scheduler | ReduceLRonPlateau(factor=0.8, patience=5, min_lr=1e-7) |
| **Inference** | |
| SDE steps $T$ | [15,25,50] |
| Guidance strength $\alpha$ | [0.05,0.06,0.07] |

*Table 5.* MD simulation setups using `OpenMM`.

| Property | Value |
|---|---|
| Forcefield | AMBER-14 |
| Integrator | LangevinMiddleIntegrator |
| Integration time step | 1fs |
| Frame spacing | 1ps |
| Friction coefficient | $0.5\mathrm{ps}^{-1}$ |
| Temperature | 300K |
| Solvation model | GB-OBC I |
| Electrostatics | CutoffNonPeriodic |
| Cutoff | 2.0nm |
| Constraints | HBonds |

*Table 6.* Pretraining dataset details.

| Domain | Dataset | # Items | Equilibrium | Off-equilibrium | Traj. | Forcefields |
|---|---|---|---|---|---|---|
| small molecule | PCQM4Mv2 | ∼3M | ✓ | ✗ | ✗ | - |
| | ANI-1x | ∼5M | ✓ | ✓ | ✗ | DFT |
| peptide | PepMD | ∼1M | ✓ | ✓ | ✓ | AMBER-14/GB-OBC I |
| protein | PDB | ∼200K | ✓ | ✗ | ✗ | - |
| | ATLAS | ∼500K | ✓ | ✓ | ✓ | AMBER-14/GB-OBC I |
| | SPF | ∼2M | ✗ | ✓ | ✗ | revPBE-D3(BJ)/def2-TZVP |

*Table 7.* Details of the trajectory datasets.

| Properties | MD17&22 | PepMD | ATLAS |
|---|---|---|---|
| Frame spacing | 0.5fs | 1ps | 10ps |
| Simulation time per traj | not fixed | 100ns | 10ns |
| Coarsened time $\tau$ | 100ps | 10ps | 100ps |
| # Training trajs | 8 | 269 | 790 |
| # Training pairs per traj | 5,000 | 5,000 | 500 |
| # Valid pairs per traj | 500 | 500 | 100 |
| # Test trajs | 5 | 14 | 14 |

can summarize two key observations:

1. Within the tested range, the validity metric improves significantly as $\alpha$ increases, while other metrics generally exhibit deteriorating trends. This suggests that the force guidance kernel enhances the comprehension of physical priors, thereby facilitating higher-quality sample generation, but may simultaneously constrain the exploration breadth of the state space to some extent.

2. As $T$ increases, most metrics show varying degrees of degradation. This is likely because excessive discretization of the SDE process leads to greater error accumulation, indicating that a small $T$ is sufficient for balancing accuracy and efficiency within the bridge matching framework.

*Table 8.* Ablation results of SDE steps $T$ and the guidance strength $\alpha$ on PepMD test set. Values of each metric are first averaged over 3 independent runs for each peptide and then shown in mean/std of all 14 test peptides.

| Hyperparameters | JS DISTANCE ($\downarrow$) | | | | VAL-CA ($\uparrow$) | CONTACT ($\downarrow$) |
|---|---|---|---|---|---|---|
| | PWD | RG | TIC | TIC-2D | | |
| $T = 15, \alpha = 0.05$ | 0.328/0.149 | 0.330/0.189 | 0.510/0.124 | 0.731/0.074 | 0.575/0.139 | 0.157/0.088 |
| $T = 15, \alpha = 0.06$ | 0.340/0.143 | 0.372/0.187 | 0.511/0.114 | 0.740/0.059 | 0.594/0.100 | 0.167/0.090 |
| $T = 15, \alpha = 0.07$ | 0.349/0.144 | 0.384/0.206 | 0.523/0.132 | 0.736/0.074 | 0.607/0.138 | 0.195/0.091 |
| $T = 25, \alpha = 0.05$ | 0.391/0.141 | 0.474/0.190 | 0.507/0.134 | 0.738/0.066 | 0.441/0.117 | 0.231/0.087 |
| $T = 25, \alpha = 0.06$ | 0.404/0.171 | 0.445/0.222 | 0.505/0.142 | 0.734/0.078 | 0.468/0.136 | 0.244/0.110 |
| $T = 25, \alpha = 0.07$ | 0.409/0.159 | 0.488/0.218 | 0.516/0.129 | 0.742/0.078 | 0.496/0.147 | 0.239/0.114 |

**Atomic Embedding Expansion** To investigate the contribution of atomic embedding expansion to pretraining, we conduct additional ablation experiments. Specifically, we remove the expanded embedding $A_e$ from the model architecture while keeping all other parameters the same as in Table 4. The model is then pretrained and finetuned on the PepMD dataset following the same procedure as described in the main text, and the ablation results on PepMD test set are shown in Table 9. It is evident that removing $A_e$ leads to a decline in nearly all evaluation metrics to varying degrees, demonstrating the effectiveness and necessity of our atomic embedding expansion technique in cross-domain scenarios.

*Table 9.* Ablation Results of atomic embedding expansion on PepMD test set. Values of each metric are shown in mean/std of all 14 test peptides. The best result for each metric is shown in **bold**.

| MODELS | JS DISTANCE ($\downarrow$) | | | | VAL-CA ($\uparrow$) | CONTACT ($\downarrow$) |
|---|---|---|---|---|---|---|
| | PWD | RG | TIC | TIC-2D | | |
| UniSim/g | **0.332**/0.135 | **0.332**/0.161 | **0.510**/0.115 | 0.738/0.064 | **0.505**/0.112 | **0.162**/0.076 |
| UniSim/g w/o $A_e$ | 0.389/0.175 | 0.453/0.233 | 0.516/0.135 | **0.732**/0.053 | 0.397/0.132 | 0.228/0.119 |

### E.2. Inference Efficiency

In this section, we compare the inference efficiency of UniSim with that of MD performed using `OpenMM`. Following Timewarp ([Klein et al., 2024](#)), we use the effective-sample-size per second of wall-clock time (ESS/s) as the evaluation metric. Figure 8 illustrates the statistical results of ESS/s between UniSim and MD on PepMD test set, demonstrating that UniSim achieves, on average, approximately 25 times higher efficiency compared to MD.

Meanwhile, since conformation refinement using `OpenMM` is performed at each inference iteration (*i.e.*, generating a new state) for peptide or protein generation, we provide the statistics of the number of optimization steps required per iteration as follows: (1) mean: 69.3, (2) median: 55, and (3) maximum: 2,075, as illustrated in Figure 8. Additionally, for each iteration, the average inference time is 0.120 s and the average optimization time is 0.152 s. Therefore, the computational overhead remains within the same order of magnitude with the refinement step.

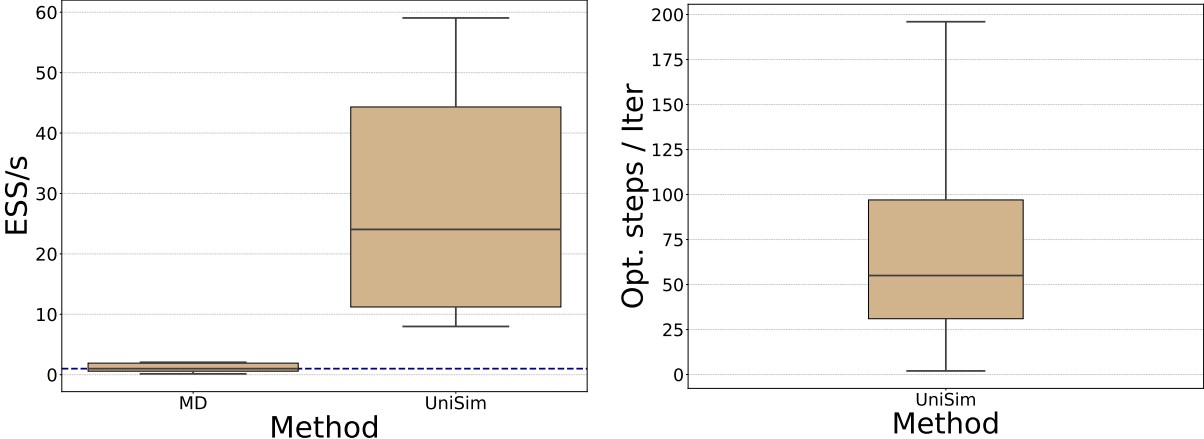

*Figure 8.* Statistical results for evaluating the model's efficiency. **Left.** The effective-sample-size per second of wall-clock time (ESS/s) on PepMD test set. For the convenience of comparison, the values were converted into ratios relative to the median of results from MD (the blue dashed line). **Right.** The optimization steps performed by `OpenMM` per iteration on PepMD test set.

## F. Computing Infrastructure

UniSim was trained on 8 NVIDIA GeForce RTX 3090 GPUs within a week. The inference procedures were performed on one NVIDIA GeForce RTX 3090 GPU.

