# OpenReview forum: "UniSim: A Unified Simulator for Time-Coarsened Dynamics of Biomolecules"
_ICML.cc/2025/Conference — ICML 2025 poster_

### Official Review · Reviewer_mTdA · 2025-02-21

**Overall Recommendation:** 2

**Summary:**

This paper introduces UniSim, a model for simulating time-coarsened molecular dynamics of small molecules, peptides, and proteins. UniSim employs a multi-stage approach: First, a unified atomic representation model is pretrained on a diverse collection of molecular datasets (both equilibrium and off-equilibrium conformations) using a multi-task approach. Then, a vector field model is trained using the stochastic interpolants framework to predict the probability transition kernel for large timesteps (time-coarsening). Finally, a force guidance kernel is trained with other model parameters frozen to adapt (fine-tune) the model to specific chemical environments and force fields. The authors evaluate UniSim on benchmark datasets for each of the three molecular domains and compare to existing methods for time-coarsened molecular dynamics.

## update after rebuttal

Please see comments below.

**Claims And Evidence:**

The main claims are:

1. UniSim is the first time-coarsened dynamics model exhibiting transferability across diverse molecular domains (small molecules, peptides, and proteins)

2. A multi-task pretraining approach learns a unified atomic representation

3. The stochastic interpolant framework and force guidance allow for rapid fine-tuning to different chemical environments, learning the state transition patterns over long time steps

Regarding the first claim, even though the authors write that their model is transferable, in practice, they train separate force guidance kernels for specific tasks. In my understanding, a truly transferable model should work "out-of-the-box" without any need for fine-tuning or output heads trained from scratch. It would thus be more accurate to say that they train a "reusable backbone", or "model that can be efficiently finetuned" (or similar formulations). Whether UniSim is the first model to achieve this level of transferability is also unclear: The authors should compare to existing models that are finetuned in a similar manner to UniSim, and then compare the performance of different finetuned backbones.

For the second claim, the "multi-task pretraining" is actually a single task (force matching) pre-training in disguise. The off-equilibrium loss L_o (Eq.7) is the standard squared error force loss that is also used to train machine-learned force fields: H_out[i] can be regarded as the atomic energy contribution of atom i to the total predicted energy sum_i H_out[i]. Then, grad_X sum_i H_out[i] just computes the gradient of this predicted energy, i.e., the associated forces. By minimizing L_o, the predicted forces are matched to the reference forces. It turns out that the equilibrium loss L_e (Eq.10) is actually the same objective in disguise (i.e., predicted forces are matched to a reference force). To see this, recall that any potential energy surface is well-approximated by a harmonic potential close to an equilibrium (take the Taylor expansion and truncate after the second-order term, the first-order term vanishes because the forces are zero at equilibrium structures, which are minima of the potential). Then, the term -(X'-X)/sigma_e is just the force of a harmonic potential given by V(X') = 1/(2 sigma_e) (X'-X)^2. The authors should make this connection more transparent and consider changing  "multi-task pretraining" to "force matching pretraining" (or a similar term), and clearly explain that L_e is just a clever trick to generate synthetic data with estimated force labels from equilibrium structures.

For the last claim, I think the shown data does not support the statement that the state transition patterns for long-time steps are learned. In fact, the TIC0/TIC1 plots shown in several figures clearly show that the sampled distributions do **not** closely approximate the probability distributions sampled during MD. Perhaps UniSim is competitive with other models that attempt time-coarsened dynamics, and I acknowledge that it is a step in the right direction, but the task is definitely not solved in a satisfactory manner.

Finally, there is another claim that is hidden in the conclusion. The authors write: "Experiments conducted on small molecules, peptides and proteins have fully verified the superiority of UniSim in distribution similarity compared to MD trajectories and transferability to out-of-distribution molecular domains." However, as I already stated above, the agreement to distributions sampled in MD trajectories looks actually quite poor. The metrics the authors look at (e.g., distance histograms, contact maps, etc.) are certainly informative, but they are not sufficient. The ultimate measure of quality must be how well actual thermodynamic observables are reproduced by the time-coarsened dynamics. After all, accelerated dynamics are useless if they give the wrong results, and the experiments that are done currently are not sufficient to "fully verify the superiority of UniSim", in particular with respect to regular MD.

**Essential References Not Discussed:**

I did not spot any essential omissions.

**Experimental Designs Or Analyses:**

The experiments that were performed and the analyses that were done seem sound to me. However, there is an aspect of the work, which should be discussed more prominently: In Appendix C.2., the authors mention that every prediction of their model is followed by an energy refinement with OpenMM to prevent error accumulation. This is a very important detail that should be discussed in the main text, and it is also necessary to state how many optimization steps need to be performed on average. Every evaluation of OpenMM used for the minimization is roughly equivalent to a single regular MD time step, so this greatly affects the actual speedup offered by UniSim. For example, if UniSim can leverage a time step that is, say, 1000 times larger than a regular MD time step, but then on average 1000 optimization steps with OpenMM are necessary to stabilize the structure, there is no net gain in terms of efficiency (on the contrary, this would be less efficient than just running regular MD).

**Methods And Evaluation Criteria:**

The proposed methods are appropriate for the problem. However, the evaluation criteria are not sufficient to show that the method works well and is practically useful. As mentioned above, to truly measure the quality of a time-coarsened dynamics, it is necessary to check how well it reproduces thermodynamic observables extracted from regular MD simulations. I think the authors should at least try a very simple example, e.g., reproducing the free energy surface of alanine-dipeptide (a well-studied small model system) with UniSim, so readers are better able to judge whether the presented model is useful for practical studies or not.

**Other Comments Or Suggestions:**

I think the TIC0/TIC1 plots could be improved: Instead of showing a scatter of the time-coarsened sampling over a contour of the true probability distribution, it would be better to do a side-by-side comparison of the true probability distribution and the distribution sampled by UniSim's time-coarsened dynamics (both as contours). This would allow a more direct comparison and allow readers to better judge the agreement of time-coarsened dynamics with the ground truth.

**Other Strengths And Weaknesses:**

**Strengths**

+ The core idea of a unified simulator for diverse biomolecules is valid.
+ The paper is well-written and easy to follow.

**Weaknesses**

- Lack of a direct comparison of computational efficiency with baselines and classical MD (especially considering that minimization with regular force fields seems to be necessary to stabilize trajectories).
- Lack of experiments that investigate the prediction of thermodynamic observables with the proposed framework, assessing the practical usefulness of the approach
- Some aspects of the work are unnecessarily "obfuscated" (e.g., the multi-task pretraining is really a single task in disguise), or hidden in the appendix (e.g., the need for minimization with a classical force field for stabilization).
- Some design choices are questionable, for example, the energy normalization described in Appendix C.1. effectively introdces a dataset-dependent unit conversion. This may not cause a large issue if all datasets are sampled at roughly comparable temperatures, but it seems unjustified from a theoretical standpoint.

**Questions For Authors:**

1. Can you provide a quantitative comparison of the computational cost (e.g., wall-clock time or speedup factor) of UniSim (including the time needed for structure refinement via minimization with OpenMM) compared to classical MD?
2. Have you performed any experiments to assess the stability and accuracy of UniSim on longer timescales? How do the cumulative prediction errors affect the results over longer simulations?
3. How sensitive is the performance of UniSim to the choice of hyperparameters? Did you perform any hyperparameter optimization, and if so, what were the key findings?
4. Could you elaborate on the motivation behind the atomic embedding expansion technique? How does it specifically help in distinguishing different chemical environments compared to using just the periodic table? Did you do ablation studies?
5. Could you elaborate a bit more on the intuition and advantage of using the "gradient-environment subgraph" approach? What is the expected impact of different choices of delta_min and delta_max?

**Relation To Broader Scientific Literature:**

The paper appears to cite the most relevant prior work. However, when discussing classical MD methods, quantum mechanics, and empirical force fields in the introduction, the provided references, while not outright wrong, are somewhat questionable. It almost makes it appear as if the cited works introduced the corresponding concepts. Perhaps it would be more appropriate to cite earlier seminal work here.

**Theoretical Claims:**

The paper includes theoretical justifications, particularly for the stochastic interpolant framework (Section 3.3 and Appendix B). I have not checked the presented proofs carefully, but at a first glance, they seem sound to me.

---

> ### Author Rebuttal · Authors · 2025-04-01
>
> - **Q1:** The first claim & its transferability
>
> Firstly, we strongly agree that the term "reusable backbone" is more suitable to describe UniSim, which will be used in our revised manuscript.
>
> Secondly, we argue that the force kernel is not strictly necessary for cross-domain generalization. Figure 4 shows that UniSim/g reproduces TIC-2D distributions for MD22 small molecules without any fine-tuning procedure. This zero-shot transfer capability confirms the learned physical representations transcend specific training domains.
>
> Lastly, we are **unable** to apply identical experimental setups to baselines for validating transferability probably due to:
>
> 1. Architectural Constraints: Methods like FBM incorporate protein-specific atomic representations (e.g., residue index) that are fundamentally incompatible with small molecules.
> 2. Pretraining Deficiency: None of them provides pretraining interfaces, preventing knowledge transfer to novel atomic species after training solely on peptide/protein data.
>
> - **Q2:** The term "multi-task pretraining"
>
> We would like to clarify that "multi-task pretraining" (line 220) does **not** refer to different loss formulations here. In particular, different "tasks" is defined as different force fields where the force labels come from (line 220-229). By using such multi-task pretraining, we address label inconsistency issues owing to varying force field parameters across different dataset. We hope the misunderstanding can be resolved.
>
> - **Q3:** The practical usefulness in longer timescales
>
> To specifically address your concerns and validate UniSim’s practical usefulness, we have performed long-timescale simulations on a well-studied system, alanine-dipeptide. Details of the experimental setup can be found in our response to **Q1,** **Reviewer KWXR**, where we show UniSim successfully reproduces the free energy landscape ([Figure 1](https://anonymous.4open.science/r/uni-sim-3BBB/rebuttal/ad_ramachandran.png)), with its stability been verified based on RMSD to the initial state ([Figure 2](https://anonymous.4open.science/r/uni-sim-3BBB/rebuttal/ad_rmsd_over_time.png)).
>
> - **Q4:** The energy minimization technique & efficiency
>
> Firstly, we emphasize that the energy minimization is **almost only** applied to hydrogen atoms with heavy atoms constrained by predefined harmonic potentials (line 682). For this reason, we placed the details in the appendix instead of the main text.
>
> Regarding the computational efficiency, we provide [Table 5](https://anonymous.4open.science/r/uni-sim-3BBB/rebuttal/ess_pepmd.png), which demonstrates that UniSim maintains a 1–2 order-of-magnitude advantage in ESS/s (used in Timewarp) over MD, with refinement step included.
>
> - **Q5:** The energy normalization trick
>
> Firstly, we claim that UniSim focuses on capturing relative energy differences rather than absolute values, as the weight of the Boltzmann constraint can be further modulated through the guidance strength $\alpha$. Secondly, we confirm that all trajectories in PepMD and ATLAS datasets were sampled at 300K, while MD17 data was collected at around 500K. Thus the temperature consistency within each dataset is guaranteed.
>
> - **Q6:** The visualization of TIC plots
>
> Given that UniSim's trajectories are relatively short (1,000 frames), contour plots failed to demonstrate clear distributions and yielded suboptimal visualization. By the way, Ramachandran plots in the same format as MD are shown in ([Figure 1](https://anonymous.4open.science/r/uni-sim-3BBB/rebuttal/ad_ramachandran.png)) for long-timescale simulations.
>
> - **Q7:** Hyperparameter sensitivity
>
> In this experiment, the primary hyperparameters we tuned were: (1) the SDE inference steps $T$ and (2) the guidance strength $\alpha$. Due to space limitations, please refer to **Q2, reviewer mRvK** for our ablation studies and further discussions.
>
> - **Q8:** Intuition of the atomic embedding expansion
>
> Here we will elaborate on the rationale behind our design of atomic embedding expansion.
>
> Since we aim for cross-domain representation learning, using fine-grained vocabulary specific to proteins would conflict with small-molecule representation. While using the periodic table is the simplest alternative, it will result in significant information loss on **regularly occurring patterns**. Therefore, we adopt the periodic table as the base vocabulary $A_b$ and extend each element with a supplementary vector of length $D$, thereby modeling recurring chemical environments while maintaining a unified representation.
>
> Specifically, we validate the effectiveness of atomic embedding expansion with the ablation on PepMD shown in [**Table 6**](https://anonymous.4open.science/r/uni-sim-3BBB/rebuttal/ablation_Ae_pepmd.png).
>
> - **Q9:** Intuition of using the "gradient-environment subgraph" approach
>
> Thank you for your valuable question. Due to space limitations, We kindly direct the reviewer to our response to **Q1, Reviewer Cnw4** for a detailed discussion.

---

> > ### Comment · Reviewer_mTdA · 2025-04-03
> >
> > I'd like to thank the authors for their replies
> >
> > > Firstly, we strongly agree that the term "reusable backbone" is more suitable to describe UniSim, which will be used in our revised manuscript.
> >
> > Thank you.
> >
> > > Lastly, we are unable to apply identical experimental setups to baselines for validating transferability probably due to:
> > > * Architectural Constraints: Methods like FBM incorporate protein-specific atomic representations (e.g., residue index) that are fundamentally incompatible with small molecules.
> > > * Pretraining Deficiency: None of them provides pretraining interfaces, preventing knowledge transfer to novel atomic species after training solely on peptide/protein data.
> >
> > I don't understand these arguments. If FBM cannot be applied to small molecules, why not apply your method to larger systems instead? And what do you mean by "none of them provides pretraining interfaces"? Why would it not be possible to pre-train on one dataset, and finetune on another? Why do you need an "interface" for that? This should be possible with any architecture.
> >
> > > We would like to clarify that "multi-task pretraining" (line 220) does not refer to different loss formulations here. In particular, different "tasks" is defined as different force fields where the force labels come from (line 220-229). By using such multi-task pretraining, we address label inconsistency issues owing to varying force field parameters across different dataset. We hope the misunderstanding can be resolved.
> >
> > In this case, I wonder even more whether the term "multi-task" is accurate here. After all, it's the same task (force prediction), you just want to reconcile the fact that different datasets use different levels of theory (and therefore have inconsistent force labels), or they don't have force labels at all (so you approximate them). A different term/description would make much clearer what is actually happening.
> >
> > > Firstly, we emphasize that the energy minimization is almost only applied to hydrogen atoms with heavy atoms constrained by predefined harmonic potentials (line 682). For this reason, we placed the details in the appendix instead of the main text.
> >
> > This does not really answer my question. Can you provide any statistics on how many optimisation steps are required on average (at most)? Also, that you almost only need to optimise hydrogen atoms does not change the fact that this seems to be necessary to make the method stable. It's an important detail that should not be hidden in the appendix.
> >
> > > Firstly, we claim that UniSim focuses on capturing relative energy differences rather than absolute values, as the weight of the Boltzmann constraint can be further modulated through the guidance strength. Secondly, we confirm that all trajectories in PepMD and ATLAS datasets were sampled at 300K, while MD17 data was collected at around 500K. Thus the temperature consistency within each dataset is guaranteed.
> >
> > I don't really understand the point the authors want to make here. I never doubted that UniSim captures relative energy differences...
> >
> > > Given that UniSim's trajectories are relatively short (1,000 frames), contour plots failed to demonstrate clear distributions and yielded suboptimal visualization. By the way, Ramachandran plots in the same format as MD are shown in (Figure 1) for long-timescale simulations.
> >
> > Thank you for this additional experiment. However, I still don't find this result particularly convincing (the agreement is not very good). For example, compare to [this paper from 2019](https://www.nature.com/articles/s41524-019-0261-5), which shows much better agreement to the ground truth.
> >
> > > Since we aim for cross-domain representation learning, using fine-grained vocabulary specific to proteins would conflict with small-molecule representation. While using the periodic table is the simplest alternative, it will result in significant information loss on regularly occurring patterns. Therefore, we adopt the periodic table as the base vocabulary
> > and extend each element with a supplementary vector of length, thereby modeling recurring chemical environments while maintaining a unified representation.
> >
> > Thank you, but again, this is not answer my question. I apologise if I was unclear: I am not wondering why you use the atomic table as the "basis" (that makes sense, obviously), I am asking why it is necessary to "extend" the vocabulary with chemical environments. What deficiencies does the model have if you use *only* elements as the vocabulary?

---

> > > ### Author Response · Authors · 2025-04-07
> > >
> > > - **Q1**: If FBM cannot be applied to small molecules, why not apply your method to larger systems instead...
> > >
> > > We sincerely appreciate your question and apologize for any confusion caused. By mentioning the lack of "pretraining interface," we meant that baselines trained only on peptide data cannot **directly** handle unseen atom types-a capability that could be enabled by cross-domain pretraining, as evidenced by UniSim/g, Figure 4.
> > >
> > > We further finetune baselines (pre-trained on PepMD) on ATLAS (excluding Timewarp, which requires unavailable atomic velocity data). As shown in these [results](https://anonymous.4open.science/r/uni-sim-3BBB/rebuttal/result_atlas.png), UniSim outperforms baselines across all metrics, proving stronger transferability.
> > >
> > > - **Q2**: I wonder even more whether the term "multi-task" is accurate here...
> > >
> > > We sincerely appreciate your suggestion. In our revised version, we will replace "multi-task" with more appropriate alternatives such as "multi-head" to avoid any misunderstandings.
> > >
> > > - **Q3**: Statistics on optimisation steps.
> > >
> > > We appreciate your valuable suggestion. Here we provide the visualization of UniSim's efficiency in [Figure](https://anonymous.4open.science/r/uni-sim-3BBB/rebuttal/efficiency_stats.png) as well as statistics of the number of optimization steps required per inference step as follows: (1) mean: 69.3, (2) median: 55 and (3) maximum: 2,075. For each inference step, the average inference time is 0.120 s and the average optimization time is 0.152 s. Therefore, the computational overhead remains within the same order of magnitude with the refinement step.
> > >
> > > Moreover, since the refinement step **does** contribute to stabilizing the generation, we agree with your suggestion and will include a discussion of the refinement procedure in the main text of our revised manuscript.
> > >
> > > - **Q4**: I don't really understand the point the authors want to make here...
> > >
> > > We sincerely apologize for any confusion caused. First, we have verified that the simulation temperature is almost the same for each dataset, which could address one of your concerns. Further, after training to fit the normalized potential, we can adjust the guidance strength $\alpha$ to approximate the unnormalized one during inference. Therefore, the normalization scheme just needs to maintain the distribution characteristics of potentials, where the min-max normalization proves to be a good choice.
> > >
> > > - **Q5**: I still don't find this result particularly convincing...
> > >
> > > We sincerely appreciate your question. First, we would like to clarify that the Ramachandran plots in [Figure](https://anonymous.4open.science/r/uni-sim-3BBB/rebuttal/ad_ramachandran.png) successfully reproduce all five metastable states of AD mentioned in [this paper](https://pubs.acs.org/doi/abs/10.1021/ct400993e). UniSim also shows a close match of representative conformations for each state as illustrated in [Figure](https://anonymous.4open.science/r/uni-sim-3BBB/rebuttal/ad_metastable.png). While complete coverage may require more sampling steps, our current results could already capture the essential conformational landscape.
> > >
> > > In addition, regarding the cited work (this paper from 2019), we highlight the following key differences from our UniSim:
> > > 1. The cited work employs a coarse-grained  (CG) molecular representation, whereas UniSim adopts an all-atom representation.
> > > 2. The cited work trains networks to replace potentials in classical MD simulations, without generating trajectories directly like UniSim.
> > > 3. The Ramachandran plots in the cited work are generated via i.i.d. sampling from a VAE decoder. In contrast, UniSim produces trajectories autoregressively, which inherently involves error accumulation and thus presents a distinct challenge.
> > > 4. The cited model requires system-specific tuning of the CG resolution parameter, which limits its generalizability.
> > >
> > > That said, we are pleased to include the discussion of this work in our revised manuscript.
> > >
> > > - **Q6**: I am not wondering why you use the atomic table as the "basis"...
> > >
> > > We apologize for not addressing your question directly in our earlier response and here we give a clearer explanation.
> > >
> > > In domains like proteins, atoms follow **regular chemical patterns** (e.g., CA, CB in residues). Wet lab experiments have shown that atoms of the same pattern have stable properties (e.g., bond lengths). Considering the constraints on bond lengths and angles, these patterns exhibit discrete characteristics, which GNNs alone are not sufficient to handle.
> > >
> > > This implies that an effective embedding approach must capture these patterns. Using only the periodic table as vocabulary would yield low-resolution representations, missing those domain-specific regularities. Instead, our solution extends each element in the vocabulary to $D$ discrete patterns and maps each atom to the most possible pattern based on its neighbors in the graph, thus simplifying the understanding of complex but highly regular structures.

---

### Official Review · Reviewer_KWXR · 2025-03-13

**Overall Recommendation:** 4

**Summary:**

The paper introduces UniSim, a deep learning framework designed to simulate biomolecular dynamics over coarse-grained time steps. The method unifies the treatment of small molecules, peptides, and proteins by first learning a unified atomic representation through multi-task pretraining on a diverse set of 3D molecular datasets. Building on this representation, the authors employ a stochastic interpolant framework to model the transition probability of molecular states over long timesteps, thus enabling time-coarsened dynamics simulation. A force guidance kernel is further introduced to adapt the model to various chemical environments and force field conditions. The approach is evaluated on several datasets (e.g., PepMD, MD17/MD22, ATLAS) and compared against SOTA methods, with experiments demonstrating competitive performance in capturing both the structural and distributional properties of MD trajectories.

**Claims And Evidence:**

- Main Claims:
The authors claim that UniSim is the first simulator capable of transferably modeling time-coarsened dynamics across diverse biomolecular domains. In addition, they assert that the multi-task pretraining yields a robust unified atomic representation, and that the incorporation of a force guidance module significantly enhances sample validity and adherence to underlying physical constraints.

- Evidence:
The claims are supported by extensive experimental results. Quantitative metrics such as Jensen–Shannon (JS) distances are reported on multiple datasets. Comparative analyses and ablation studies (e.g., UniSim with and without force guidance) further corroborate the improvements brought by the proposed components. While the evidence is comprehensive, the paper also acknowledges limitations (e.g., relatively short generated trajectories and suboptimal validity in protein simulations), which helps contextualize the claims.

**Essential References Not Discussed:**

This paper is a good summary of this field.

**Experimental Designs Or Analyses:**

- Design:
The experimental design is extensive and multifaceted. The authors conduct evaluations across three molecular domains (small molecules, peptides, proteins) using multiple datasets. They perform ablation studies comparing the full UniSim model with a variant that omits the force guidance module (UniSim/g) to isolate its effect.

- Analyses:
Quantitative comparisons against competitive baseline methods (FBM, Timewarp, ITO, Score Dynamics) are provided via tables and figures that report JS distances, validity scores, and contact errors. Visualizations (e.g., TIC plots, contact maps) illustrate the capability of UniSim to capture the free energy landscape and the transitions between metastable states.

- Potential Issues:
Although the experiments are comprehensive, the generated trajectories are relatively short, which may limit insights into long-term dynamics. So adding experiments about the exploration of with conformational space, such as the experiments in F$^3$low, can further illustrate the point.

**Methods And Evaluation Criteria:**

- Methodology: The approach comprises three main stages:
1. Unified Pretraining: A multi-task pretraining strategy is used to learn an atomic representation from diverse 3D molecular datasets. An SO(3)-equivariant GNN is employed along with novel techniques like gradient-environment subgraph construction and atomic embedding expansion.
2 .Vector Field Model: A stochastic interpolant framework is used to fit the probability transition kernel of MD trajectories over a coarse time step. This involves training neural networks to predict both a drift term and a noise term, thereby defining an SDE that governs state evolution.
3. Force Guidance Module: To adapt to different chemical environments, a force guidance kernel is introduced. This module adjusts the learned transition dynamics using “virtual” forces derived from MD potential labels.
- Evaluation Criteria:
The method is evaluated using metrics that assess both the physical plausibility (e.g., VAL-CA, contact map errors) and the distributional similarity (JS distances on various projections) of generated trajectories relative to ground-truth MD data. These criteria are well chosen for the problem of capturing the complex free energy landscapes and metastable states in biomolecular dynamics.

**Other Comments Or Suggestions:**

No other comments or suggestions.

**Other Strengths And Weaknesses:**

I think the concepts presented in this paper are not novel, but it is a very solid piece of work that blends the appropriate concepts more seamlessly and does to SOTA on the various task metrics, which is very helpful to the research community

**Questions For Authors:**

I don't have a specific question.

**Relation To Broader Scientific Literature:**

This paper combines pre-training methods, stochastic interpolation methods in generative modeling, and the classifer-guided approach that is popular in generative models.

**Theoretical Claims:**

The paper states theoretical propositions (e.g., Proposition 3.1) that connect the learned vector field model with the force guidance adjustments. The proofs—provided in the appendix—are built on established frameworks from stochastic interpolants and score matching techniques. They derive closed-form relations between the drift and noise components of the SDE under both the baseline and force-guided settings. Although this is not novel, giving proof enhances the completeness of the paper.

---

> ### Author Rebuttal · Authors · 2025-03-31
>
> - **Q1**: Potential Issues: Although the experiments are comprehensive, the generated trajectories are relatively short, which may limit insights into long-term dynamics. So adding experiments about the exploration of with conformational space, such as the experiments in F3low, can further illustrate the point.
>
> Thank you for your instructful suggestion! To further validate the stability and applicability of UniSim in long-timescale simulations, we conducted additional experiments on a well-studied molecular system, alanine dipeptide (AD), consisting of only 22 atoms while exhibiting comprehensive free energy landscapes.
>
> Specifically, following the task setup in Timewarp [1], we attempt to finetune our vector field model trained on PepMD (i.e., UniSim/g) to AD before performing long-timescale simulations. Firstly, we obtained three **independently** sampled MD trajectories of alanine dipeptide (AD) with the simulation time of 250 ns from [mdshare](https://markovmodel.github.io/mdshare/ALA2/#alanine-dipeptide), which were assigned as the training/validation/test set. The coarsened timestep $\tau$ was set to 100 ps, and 200,000 data pairs were sampled from the training and validation set, respectively. Then UniSim/g was finetuned on AD with the learning rate of 1e-4 for at most 300 epochs, after which an force guidance kernel was trained based on the finetuned model with the learning rate of 5e-4.
>
> After we obtain the best checkpoint of UniSim evaluated on the validation set, we perform long-timescale simulations for a chain length of 100,000 to explore the metastable states of AD. We show the Ramachandran and TIC-2D plots of UniSim and the test MD trajectory in [Figure 1](https://anonymous.4open.science/r/uni-sim-3BBB/rebuttal/ad_ramachandran.png). Building upon previous research [2], UniSim has demonstrated robust performance in long-timescale simulations by effectively exploring key metastable states of AD, including C$5$, C$7_{eq}$, $\alpha_{R}'$, $\alpha_{R}$ as well as $\alpha_{L}$. Moreover, the relative weights of generated conformation ensembles across different metastable states show good agreement with MD, confirming that UniSim accurately reproduces AD's free energy landscape in long-timescale simulations.
>
> Furthermore, to investigate the stability of UniSim in long-timescale simulations, we selected the root-mean-square deviation (RMSD) of heavy atoms relative to the initial state as the validation metric. The plots of RMSD over time compared to the MD trajectory are presented in [Figure 2](https://anonymous.4open.science/r/uni-sim-3BBB/rebuttal/ad_rmsd_over_time.png). As evident from the figure, the RMSD variations and ranges of the trajectory generated by UniSim exhibit strong consistency with that of MD, demonstrating the model's stability in long-timescale simulations.
>
> **Reference**
> > [1] Klein, L., Foong, A., Fjelde, T., Mlodozeniec, B., Brockschmidt, M., Nowozin, S., ... & Tomioka, R. (2023). Timewarp: Transferable acceleration of molecular dynamics by learning time-coarsened dynamics. Advances in Neural Information Processing Systems, 36, 52863-52883.
>
> > [2] Wang, H., Schütte, C., Ciccotti, G., & Delle Site, L. (2014). Exploring the conformational dynamics of alanine dipeptide in solution subjected to an external electric field: A nonequilibrium molecular dynamics simulation. Journal of Chemical Theory and Computation, 10(4), 1376-1386.

---

> > ### Comment · Reviewer_KWXR · 2025-04-05
> >
> > I thank the author for conducting more experiments, my concerns were largely resolved and I think this paper deserves a 4 out of 5 and I will keep my score.

---

> > > ### Author Response · Authors · 2025-04-09
> > >
> > > We thank the reviewer for the positive evaluation and valuable comments, which have helped us strengthen the experiments and improve our manuscript. We sincerely appreciate your time and insightful feedback!

---

### Official Review · Reviewer_Cnw4 · 2025-03-15

**Overall Recommendation:** 3

**Summary:**

The paper introduces UniSim, a deep learning-based unified simulator for time-coarsened MD simulation. The framework aims to improve the transferability and efficiency of long-timescale molecular simulations across different biomolecular domains (small molecules, peptides, and proteins), and consistes of a pretraining module with unified atomic representation, a state transition module using stochastic interpolant , and optional force guidance kernel. Evaluations on small molecules, peptides, and proteins show competitive results w.r.t. several baselines.

**Claims And Evidence:**

Mostly yes, the novelty  of combining atomistic pretraining with stochastic-interpolant-based dynamics modeling is backed by previous literature. The effectiveness of the force guidance kernel is supported by ablation, and cross-domain transferability is shown on datasets such as MD22.

**Essential References Not Discussed:**

Not that I found.

**Experimental Designs Or Analyses:**

The designs are largely consistent with the claims, see above.

**Methods And Evaluation Criteria:**

Yes, see above.

**Other Comments Or Suggestions:**

See above.

**Other Strengths And Weaknesses:**

Strengths
1. The pretraining + stochastic interpolant framework is novel.
2. Force guidance kernel further improves flexibility in different chemical environments.
3. Results showing good cross-domain transferability.

 Weaknesses:
1. Writing clarity in the methods needs improvement. For example, the gradient-environment subgraph part was a bit confusing and needs some more intuition on the design.
2. Effect of predefined timestep τ on model accuracy is unexplored. Interesting to discuss how τ would affect model performance. No major experiment is required, but would like at least some discussion on the intuition.

**Questions For Authors:**

See above.

**Relation To Broader Scientific Literature:**

The work enables a broader, cross-domain application of time-coarsen MD simulation and also promotes the combination of pretraining with dynamic modelling.

**Theoretical Claims:**

Proofs are included in the supplement which I didn't check.

---

> ### Author Rebuttal · Authors · 2025-03-31
>
> - **Q1:** Writing clarity in the methods needs improvement. For example, the gradient-environment subgraph part was a bit confusing and needs some more intuition on the design.
>
> Thanks for the question. We will elaborate on the design rationale of the gradient-environment subgraph as clearly as possible.
>
> **Scale Invariance:** A crucial challenge in training unified representation models arises from the vast scale discrepancy between molecular systems: small molecules typically contain ~10^1 atoms, while proteins often comprise ~10^3 or more atoms. Conducting direct full-atom representation learning without proper processing could lead to non-transferrable representations across different molecular domains.
>
> **Physical Faithfulness:** We notice that, from Newtonian mechanics, atomic motion is governed by the forces acting on each atom. Crucially, long-range intermolecular forces (e.g., van der Waals interactions) decay exponentially with distance, becoming negligible beyond ~10 Å empirically. This implies that the force acting on any atom predominantly originates from its local environment within a 10 Å radius sphere. This physical insight motivates our design of the environment subgraph paradigm, where atomic force computation can be localized to such spherical subgraphs. This approach makes it feasible to decompose large biomolecules into manageable subgraphs for training.
>
> **Computational Efficiency:** Furthermore, during training, we hope that the number of atoms that contribute to loss computation should be the same order of magnitude as that of small molecules or peptides (10^1~10^2 atoms). We denote those atoms as $G_g$. From the above, the complete force-determining environment for $G_g$ is the union of the spherical subgraphs centered on each atom in $G_g$ with radius of around 10 Å, denoted as $G_e$. By using $G_e$ as the training input, we effectively constrain the number of atoms involved in gradient computation to $|G_g|$, which aligns with our intention. Specifically, for the sake of convenience, we adopt Eqs. (2-3) for constructing training data, thus avoiding computing the union of hundrends of point sets.
>
> Accordingly, our hyperparameters should satisfy:
>
> - Scale Matching: $G_g$ contains $10^1$-$10^2$ atoms (aligning with small molecule sizes)
> - Physical Consistency: $\delta$\_max - $\delta$\_min ≥ 10 Å ensures force computation completeness
>
> Moreover, regarding the impact of the selection of different $\delta$\_min and $\delta$\_max, we argue that the choice of $\delta$\_min can be arbitrary as long as the spherical region with radius delta_min contains an approximately appropriate number of atoms. In contrast, the selection of $\delta$\_max must be grounded in physical priors. Crucially, if $\delta$\_max - $\delta$\_min is so small that it shields some strong interactions, it will inevitably degrade model performance. On the contrary, once $\delta$_max - $\delta$\_min exceeds a certain threshold (e.g., 10 Å), further increasing $\delta$\_max yields negligible benefits.
>
> - **Q2:** Effect of predefined timestep τ on model accuracy is unexplored. Interesting to discuss how τ would affect model performance. No major experiment is required, but would like at least some discussion on the intuition.
>
> Thanks for the question. Regarding the impact of $\tau$ on the model performance, we provide the following explanations:
>
> - If $\tau$ is extremely small (comparable to the MD integral timestep), the conformations of $(x_0,x_1)$ would be very similar. While this may intuitively improve model accuracy, the efficiency could become comparable to or even worse than classical MD, contradicting our goal of accelerating MD simulations.
> - If $\tau$ falls within a reasonable range, increasing $\tau$ reduces the time correlation between $(x_0,x_1)$, making the learning task more challenging but enabling the model to explore more state space with a shorter simulation.
> - If $\tau$ exceeds a certain threshold, $x_0$ and $x_1$ can be considered as independent samples from the Boltzmann distribution, rendering the model incapable of learning dynamic transition features between states.
>
> In conclusion, we believe that a moderate $\tau$ is a reasonable choice.
>
> **Reference**
>
> > [1] Kong, X., Huang, W., & Liu, Y. Generalist Equivariant Transformer Towards 3D Molecular Interaction Learning. In *Forty-first International Conference on Machine Learning*.

---

### Official Review · Reviewer_mRvK · 2025-03-21

**Overall Recommendation:** 3

**Summary:**

The paper presents UniSim, a unified simulator for time‐coarsened dynamics of biomolecules. It proposes a multi-task pretraining method to learn a unified atomic representation across small molecules, peptides, and proteins. The simulator uses a stochastic interpolant framework to learn long-timestep state transitions and introduces a force guidance module to adapt the generated trajectories to different chemical environments. Experiments are reported on peptides, small molecules, and proteins, with comparisons against baselines such as FBM, Timewarp, ITO, and Score Dynamics.

**Claims And Evidence:**

The paper claims that UniSim is the first unified model for time-coarsened dynamics across diverse biomolecular domains and that its multi-task pretraining and force guidance improve distributional similarity and validity. While the theoretical formulations are supported by derivations, the experimental evidence is less convincing. In particular, the validity metrics for protein simulations remain unsatisfactory, and the improvements over existing methods are modest. These factors weaken the claims regarding robust performance and cross-domain transfer.

**Essential References Not Discussed:**

While the paper covers most key works, it would benefit from a brief discussion of methods that integrate reinforcement learning or adaptive sampling techniques for MD acceleration, as well as approaches from enhanced sampling (e.g., metadynamics) that address similar problems in different ways.

**Experimental Designs Or Analyses:**

The experiments span multiple molecular domains with comparisons against several baselines. The design is standard, and the benchmarks are appropriate. However, the results—particularly for proteins—do not show significant improvements in key validity metrics. The lack of rigorous statistical analysis (e.g., confidence intervals) and limited ablation studies on the effect of force guidance and timestep choices reduce confidence in the claimed advantages.

**Methods And Evaluation Criteria:**

The proposed methods include:
- A unified atomic representation using an SO(3)-equivariant GNN.
- A stochastic interpolant-based generative framework to bridge long simulation timesteps.
- A force guidance module to enforce Boltzmann-like behavior during sampling.

The evaluation criteria (JS distances, VAL-CA, CONTACT, etc.) are standard and appropriate for MD simulations. However, some aspects (e.g., sensitivity to hyperparameters and error accumulation over long rollouts) are not thoroughly explored.

**Other Comments Or Suggestions:**

N/A

**Other Strengths And Weaknesses:**

N/A

**Questions For Authors:**

1. Statistical Significance: Can the authors provide statistical analysis (e.g., confidence intervals or p-values) for the reported improvements, especially in the protein validity metrics?
2. Hyperparameter Sensitivity: How does the choice of the coarse timestep ($\tau$) affect the simulation accuracy and validity? Have experiments been conducted to assess sensitivity to this parameter?
3. Training Data Balance: Was the unified model trained on a balanced dataset across small molecules, peptides, and proteins? If not, could the imbalance affect the reported transferability claims?

**Relation To Broader Scientific Literature:**

The work is positioned well within the recent literature on deep learning for MD simulations. It builds on approaches such as FBM, Timewarp, ITO, and Score Dynamics, aiming to integrate aspects from each—namely, unified representations, time-coarsened simulation, and force-guided sampling. The paper accurately cites recent advances and situates its contributions relative to them.

**Theoretical Claims:**

The paper presents a proof (Proposition 3.1) linking the stochastic interpolant framework with the modified dynamics when force guidance is applied. A cursory check suggests the derivations are consistent. No major errors were detected in the presented theoretical claims, though a detailed line-by-line verification is challenging within the review context.

---

> ### Author Rebuttal · Authors · 2025-03-31
>
> - **Q1**: The validity metrics for protein simulations remain unsatisfactory.
>
> We claim that the unsatisfactory validity metric of ATLAS stems from two primary factors:
>
> 1. We pursue a unified modeling framework across small molecules and proteins, which necessitates certain compromises in protein-specific modeling (e.g., MSA).
>
> 2. The number of parameters matters: we conduct the ablation analysis by increasing the hidden dimension $H$ from 128 to 256. As demonstrated in [Table 1](https://anonymous.4open.science/r/uni-sim-3BBB/rebuttal/ablation_hidden_dim_pepmd.png) and [Table 2](https://anonymous.4open.science/r/uni-sim-3BBB/rebuttal/ablation_hidden_dim_atlas.png), the modification yields substantial improvements in validity metrics across both the PepMD and ATLAS test sets, suggesting that parameter scaling can effectively address certain performance deficiencies.
>
> - **Q2:** Some evaluation criteria are not thoroughly explored.
>
> We sincerely appreciate your valuable suggestions regarding the evaluation criteria. We address your concerns through the following systematic investigations:
>
> 1. **Hyperparameter Sensitivity**: we have now conducted the ablation study for the inference step parameter $T$ and the guidance strength $\alpha$ on the PepMD test set, which are shown in [Table 3](https://anonymous.4open.science/r/uni-sim-3BBB/rebuttal/ablation_T_alpha_pepmd.png). From the table, we can summarize two key observations:
>
>    - The validity metric improves as $\alpha$ increases, while other metrics generally exhibit deteriorating trends. This suggests that the force guidance kernel enhances the comprehension of physical priors, but may constrain the exploration of the state space to some extent.
>
>    - As $T$ increases, most metrics show degrading trends. This is likely because excessive discretization of SDE leads to greater error accumulation.
>
> 2. **Error accumulation over long rollouts**: we have performed long-timescale simulations (100,000 rollouts) on a well-studied system, alanine-dipeptide, where the detailed setup is elaborated in our response to **Q1,** **Reviewer KWXR**.  The stability of UniSim in long-timescale simulations has been verified by the alignment of RMSD relative to the initial state ([Figure 2](https://anonymous.4open.science/r/uni-sim-3BBB/rebuttal/ad_rmsd_over_time.png)).
>
> - **Q3:** The lack of rigorous statistical analysis.
>
> We have now updated the statistical analysis for protein validity metrics ([Table 4](https://anonymous.4open.science/r/uni-sim-3BBB/rebuttal/statistical_analysis.png)), which provides rigorous statistical validation of our performance claims.
>
> - **Q4**: a brief discussion of methods that integrate reinforcement learning/enhanced sampling/adaptive sampling.
>
> In response to your valuable feedback, we will further enhance the Introduction section with a brief discussion on those methods.
>
> - **Q5**: Regarding the choice of the coarse timestep $\tau$.
>
> Thanks for the instructful question. We provide the following responses:
>
> 1. First, we did not conduct ablation experiments on the coarsened timestep $\tau$. That's because selecting different $\tau$ values would require retraining the model on entirely new datasets, which is not worthwhile.
> 2. Second, we explain our methodology for selecting $\tau$ across different datasets. According to Timewarp [1] where $\tau$ was set to 500 ps, we opted for values of $\tau$ in the same order of magnitude. We then select an appropriate $\tau$ to ensure the variance of the distance between training data pairs was close to 1, thereby avoiding numerical instability during training.
> 3. Finally, we briefly discuss the impact of different $\tau$ on the performance.
>    1. If $\tau$ is extremely small, the conformations of $(x_0,x_1)$ would be very similar. While this may intuitively improve model accuracy, the sampling efficiency may become the bottleneck instead.
>    2. If $\tau$ falls within a reasonable range, increasing $\tau$ reduces the time correlation between $(x_0,x_1)$, making the learning task more challenging but probably enhances the exploration of the state space.
>    3. If $\tau$ exceeds a certain threshold, $x_0$ and $x_1$ can be considered as independent samples from the Boltzmann distribution, rendering the model incapable of learning dynamic transition features between states.
>
> In conclusion, we believe that a moderate $\tau$ is a reasonable choice.
>
> - **Q6**: Regarding the training Data Balance.
>
> We employed several engineering tricks to ensure the pretraining dataset remains as balanced as possible:
>
> - During training, we require molecules of each batch originate from the same dataset.
> - We ensure that the total number of atoms per molecule in a batch is similar using dynamic batch size.
> - We impose the same upper limit on the number of batches for each dataset in a single epoch.
>
> This approach ensures that the total number of atoms contributed by each dataset during a single epoch remains approximately balanced.

---

### Decision · Program_Chairs · 2025-05-01

**Decision:**

Accept (poster)

**Comment:**

This paper proposes a unified simulator for time‐coarsened dynamics of biomolecules, based on stochastic interpolant and force guided bridge matching to adapt the generated trajectories to different chemical environments.

The reviewers generally appreciate the proposed idea and the experiments, Reviewer mRvK and mTdA expressed concerns on metrics and evaluation procedures of the method, i.e., it is not clear whether if the proposed method actually solves the considered problem better than classic MD simulations or prior work like Wang et al., 2019. Reviewer mTdA also expressed some concerns on presentation of the method.

However, the reviewers generally agree that this work provides a solid improvement for coarsed-graining of MD for accelerated simulation. The authors also provide good effort to address the reviewer comments. I recommend acceptance for the paper, but hope the authors further improve their method through further experiments and incorporation of the reviewer comments.